# Integrating taxonomic, functional, and strain-level profiling of diverse microbial communities with bioBakery 3

Francesco Beghini[1†], Lauren J McIver[2†], Aitor Blanco-Míguez[1], Leonard Dubois[1], Francesco Asnicar[1], Sagun Maharjan[2,3], Ana Mailyan[2,3], Paolo Manghi[1], Matthias Scholz[4], Andrew Maltez Thomas[1], Mireia Valles-Colomer[1], George Weingart[2,3], Yancong Zhang[2,3], Moreno Zolfo[1], Curtis Huttenhower[2,3]*, Eric A Franzosa[2,3]*, Nicola Segata[1,5]*

[1]Department CIBIO, University of Trento, Trento, Italy; [2]Harvard T.H. Chan School of Public Health, Boston, United States; [3]The Broad Institute of MIT and Harvard, Cambridge, United States; [4]Department of Food Quality and Nutrition, Research and Innovation Center, Edmund Mach Foundation, San Michele all'Adige, Italy; [5]IEO, European Institute of Oncology IRCCS, Milan, Italy

*For correspondence:
chuttenh@hsph.harvard.edu (CH);
franzosa@hsph.harvard.edu (EAF);
nicola.segata@unitn.it (NS)

†These authors contributed
equally to this work

Competing interests: The
authors declare that no
competing interests exist.

Reviewing editor: Peter
Turnbaugh, University of
California, San Francisco, United
States

**Abstract** Culture-independent analyses of microbial communities have progressed dramatically in the last decade, particularly due to advances in methods for biological profiling via shotgun metagenomics. Opportunities for improvement continue to accelerate, with greater access to multi-omics, microbial reference genomes, and strain-level diversity. To leverage these, we present bioBakery 3, a set of integrated, improved methods for taxonomic, strain-level, functional, and phylogenetic profiling of metagenomes newly developed to build on the largest set of reference sequences now available. Compared to current alternatives, MetaPhlAn 3 increases the accuracy of taxonomic profiling, and HUMAnN 3 improves that of functional potential and activity. These methods detected novel disease-microbiome links in applications to CRC (1262 metagenomes) and IBD (1635 metagenomes and 817 metatranscriptomes). Strain-level profiling of an additional 4077 metagenomes with StrainPhlAn 3 and PanPhlAn 3 unraveled the phylogenetic and functional structure of the common gut microbe *Ruminococcus bromii*, previously described by only 15 isolate genomes. With open-source implementations and cloud-deployable reproducible workflows, the bioBakery 3 platform can help researchers deepen the resolution, scale, and accuracy of multi-omic profiling for microbial community studies.

## Introduction

Studies of microbial community biology continue to be enriched by the growth of culture-independent sequencing and high-throughput isolate genomics (*Almeida et al., 2021*; *Almeida et al., 2019*; *Forster et al., 2019*; *Parks et al., 2017*; *Pasolli et al., 2019*; *Poyet et al., 2019*; *Zou et al., 2019*). Shotgun metagenomic and metatranscriptomic (i.e. 'meta-omic') measurements can be used to address an increasing range of questions as diverse as the transmission and evolution of strains in situ (*Asnicar et al., 2017*; *Ferretti et al., 2018*; *Truong et al., 2017*; *Yassour et al., 2018*), the mechanisms of multi-organism biochemical responses in the environment (*Unified Microbiome Initiative Consortium et al., 2015*; *Blaser et al., 2016*), or the epidemiology of the human microbiome for biomarkers and therapy (*Gopalakrishnan et al., 2018*; *Le Chatelier et al., 2013*; *Thomas et al., 2019*; *Zeller et al., 2014*). Using such analyses for accurate discovery, however, requires efficient ways to integrate hundreds of thousands of (potentially fragmentary) isolate genomes with community profiles to detect novel species and strains, non-bacterial community members, microbial

phylogeny and evolution, and biochemical and molecular signaling mechanisms. Correspondingly, this computational challenge has necessitated the continued development of platforms for the detailed functional interpretation of microbial communities.

The past decade of metagenomics has seen remarkable growth both in terms of biology accessible via high-throughput sequencing and in terms of methods for doing so. Beginning with the now-classic questions of 'who's there?' and 'what are they doing?' in microbial ecology (*Human Microbiome Project Consortium, 2012*), shotgun metagenomics provide a complementary means of taxonomic profiling to amplicon-based (e.g. 16S rRNA gene) sequencing, as well as functional profiling of genes or biochemical pathways (*Morgan et al., 2013*; *Quince et al., 2017*; *Segata et al., 2013*). More recently, metagenomic functional profiles have been joined by metatranscriptomics to also capture community regulation of gene expression (*IBDMDB Investigators et al., 2019*). Methods have been developed to focus on all variants of particular taxa of interest within a set of communities (*Pasolli et al., 2019*; *Truong et al., 2017*), to discover new variants of gene families or biochemical activities (*Franzosa et al., 2018*; *Kaminski et al., 2015*), or to link the presence and evolution of closely related strains within or between communities over time, space, and around the globe (*Beghini et al., 2017*; *Karcher et al., 2020*; *Tett et al., 2019*). Critically, all these analyses (and the use of the word 'microbiome' throughout this manuscript) are equally applicable to both bacterial and non-bacterial community members (e.g. viruses and eukaryotes) (*Beghini et al., 2017*; *Olm et al., 2019*; *Yutin et al., 2018*). Finally, although not addressed in depth by this study, shotgun meta-omics have increasingly also been combined with other community profiling techniques such as metabolomics (*Heinken et al., 2019*; *Lloyd-Price et al., 2017*; *Sun et al., 2018*) and proteomics (*Xiong et al., 2015*) to provide richer pictures of microbial community membership, function, and ecology.

Methods enabling such analyses of meta-omic sequencing have developed in roughly two complementary types, either relying on metagenomic assembly or using largely assembly-independent, reference-based approaches (*Quince et al., 2017*). The latter is especially supported by the corresponding growth of fragmentary, draft, and finished microbial isolate genomes, and their consistent annotation and clustering into genome groups and pan-genomes (*Almeida et al., 2021*; *Almeida et al., 2019*; *Pasolli et al., 2019*). Most such methods focus on addressing a single profiling task within (most often) metagenomes, such as taxonomic profiling (*Lu et al., 2017*; *Milanese et al., 2019*; *Truong et al., 2015*; *Wood et al., 2019*), strain identification (*Luo et al., 2015*; *Nayfach et al., 2016*; *Scholz et al., 2016*; *Truong et al., 2017*), or functional profiling (*Franzosa et al., 2018*; *Kaminski et al., 2015*; *Nayfach et al., 2015*; *Nazeen et al., 2020*). In a few cases, platforms such as bioBakery (*McIver et al., 2018*), QIIME 2 (*Bolyen et al., 2019*), or MEGAN (*Mitra et al., 2011*) integrate several such methods within an overarching environment. While not a primary focus of this study, metagenomic assembly methods enabling the former types of analyses, including novel organism discovery or gene cataloging (*Lesker et al., 2020*; *Stewart et al., 2019*), have also advanced tremendously (*Li et al., 2015*; *Nurk et al., 2017*). Assembly-based analyses are now reaching a point of integrating microbial communities and isolate genomics as well, particularly for phylogeny (*Asnicar et al., 2020*; *Zhu et al., 2019*). These efforts have also led to increased consistency in microbial systematics and phylogeny, facilitating the types of automated, high-throughput analyses necessary when manual curation cannot keep up with such rapid growth (*Asnicar et al., 2020*; *Chaumeil et al., 2019*).

Here, to further increase the scope of feasible microbial community studies, we introduce a suite of updated and expanded computational methods in a new version of the bioBakery platform. bioBakery 3 includes updated sequence-level quality control and contaminant depletion guidelines (KneadData), MetaPhlAn 3 for taxonomic profiling, HUMAnN 3 for functional profiling, StrainPhlAn 3 and PanPhlAn 3 for nucleotide- and gene-variant-based strain profiling, and PhyloPhlAn 3 for phylogenetic placement and putative taxonomic assignment of new assemblies (metagenomic or isolate). Most of these tools leverage an updated ChocoPhlAn 3 database of systematically organized and annotated microbial genomes and gene family clusters, newly derived from UniProt/UniRef (*Suzek et al., 2007*) and NCBI (*NCBI Resource Coordinators and Coordinators, 2014*). Our quantitative evaluations show each individual tool to be more accurate and, typically, more efficient than its previous version and other comparable methods, increasing sensitivity and specificity by sometimes more than twofold (e.g. in non-human-associated microbial communities). Biomarker identifications in 1262 colorectal cancer (CRC) metagenomes, 1635 inflammatory bowel disease (IBD)

metagenomes, and 817 metatranscriptomes show both the platform's efficiency and its ability to detect hundreds of species and thousands of gene families not previously profiled. Finally, in 4077 human gut metagenomes containing *Ruminococcus bromii*, the bioBakery 3 platform permits an initial integration of assembly- and reference-based metagenomics, discovering a novel biogeographical and functional structure within the clade's evolution and global distribution. All components are available as open-source implementations with documentation, source code, and workflows enabling provenance, reproducibility, and local or cloud deployment at http://segatalab.cibio.unitn.it/tools/biobakery and http://huttenhower.sph.harvard.edu/biobakery.

## Results

bioBakery provides a complete meta-omic tool suite and analysis environment, including methods for individual meta-omic (and other microbial community) processing steps, downstream statistics, integrated reproducible workflows, standardized packaging and documentation via open-source repositories (GitHub, Conda, PyPI, and R/Bioconductor), grid- and cloud-deployable images (AWS, GCP, and Docker), online training material and demonstration data, and a public community support forum. For any sample set, quality control, taxonomic profiling, functional profiling, strain profiling, and resulting data products and reports can all be generated with a single workflow, while maintaining version control and provenance logging. All of the methods themselves, the associated training material, quality control using KneadData, and packaging for distribution and use have been updated in this version. For example, Docker images have been scaled down in size to optimize use in cloud environments, and workflows have been ported to AWS (Amazon Web Services) Batch and Terra/Cromwell (Google Compute Engine) to reduce costs through the use of spot and pre-emptive instances, respectively. All base images and dependencies have been updated as well, including the most recent Python (v3.7+) and R (v4.0+, see Materials and methods). New and updated documentation of all tools, including detailed instructions on installation in different environments and package managers, is available at http://huttenhower.sph.harvard.edu/biobakery.

### High-quality reference sequences for improved meta-omic profiling

The majority of methods within the bioBakery 3 suite leverage a newly updated reference genome and gene cataloging procedure, the results of which are packaged as ChocoPhlAn 3 (*Figure 1A*). ChocoPhlAn uses publicly available genomes and standardized gene calls and gene families to generate markers for taxonomic and strain-level profiling of metagenomes with MetaPhlAn 3, Strain-PhlAn 3, and PanPhlAn 3, phylogenetic profiling of genomes and MAGs with PhyloPhlAn 3, and functional profiling of metagenomes with HUMAnN 3.

ChocoPhlAn 3 is based on a genomic repository of 99.2 k high-quality, fully annotated reference microbial genomes from 16.8 k species available in the UniProt Proteomes portal as of January 2019 (*The UniProt Consortium, 2019*) and the corresponding functionally-annotated 87.3M UniRef90 gene families (*Suzek et al., 2015*). From this resource, ChocoPhlAn initially generates annotated species-level pangenomes associating each microbial species with its sequenced genomes and repertoire of UniRef-based gene (nucleotide) and protein (amino acid sequence) families. These pangenomes provide a uniform shared resource for subsequent profiling across bioBakery 3. HUMAnN 3 and PanPhlAn 3 are directly based on complete pangenomes for overall functional and strain profiling, whereas other tools use additional information annotated onto the catalog. PhyloPhlAn 3 focuses on the subset of conserved core gene families (i.e. present in almost all strains of a species) for inferring accurate phylogenies, and both MetaPhlAn 3 and StrainPhlAn 3 further refine core gene families into species-specific unique gene families to generate unambiguous markers for metagenomic species identification and strain-level genetic characterization.

### MetaPhlAn 3 increases the accuracy of quantitative taxonomic profiling

MetaPhlAn estimates the relative abundance of microbial taxa in a metagenome using the coverage of clade-specific marker genes (*Segata et al., 2012*; *Truong et al., 2015*). Such marker genes are chosen so that essentially all of the strains in a clade (species or otherwise) possess such genes, and at the same time no other clade contains homologs close enough to incorrectly map metagenomic reads. MetaPhlAn 3 incorporates 13.5 k species (more than twice as much as MetaPhlAn 2) with a completely new set of 1.1M marker genes (84 ± 47 mean ± SD markers per species) selected by

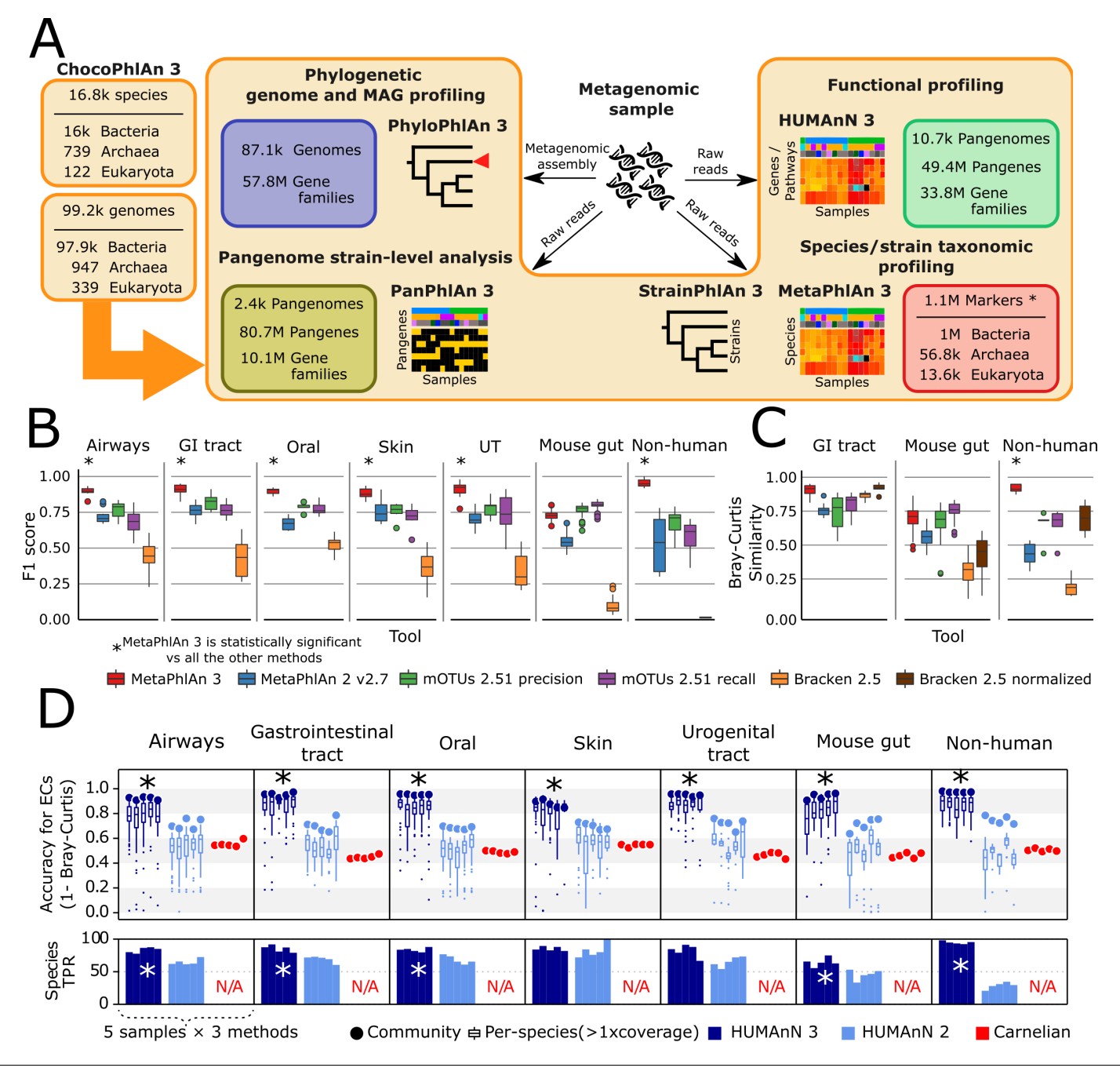

**Figure 1.** bioBakery 3 includes new microbial community profiling approaches that outperform previous versions and current methods. (**A**) The newly developed ChocoPhlAn 3 consolidates, quality controls, and annotates isolate-derived reference sequences to enable metagenomic profiling in subsequent bioBakery methods. (*The 1.1M MetaPhlAn 3 markers also encompass 61.8 k viral markers from MetaPhlAn 2 *Truong et al., 2015*) (**B**) MetaPhlAn 3 was applied to a set of 113 total evaluation datasets provided by CAMI (*Fritz et al., 2019*) representing diverse human-associated microbiomes and five datasets of non-human-associated microbiomes (*Supplementary file 1*). MetaPhlAn 3 showed increased performance compared with the previous version MetaPhlAn 2 (*Truong et al., 2015*), mOTUs2 (*Milanese et al., 2019*), and Bracken 2.5 (*Lu et al., 2017*). We report here the F1 scores (harmonic mean of the species-level precision and recall, see *Figure 1—figure supplement 1* for other evaluation scores). (**C**) MetaPhlAn 3 better recapitulates relative abundance profiles both from human and murine gastrointestinal metagenomes as well from non-human-associated communities compared to the other currently available tools (full results in *Figure 1—figure supplement 1*). Bracken is reported both using its original estimates based on the fraction of reads assigned to each taxon and after re-normalizing them using the genome lengths of the taxa in the gold standard to match the taxa abundance estimate of the other tools. (**D**) Compared with HUMAnN 2 (*Franzosa et al., 2018*) and Carnelian (*Nazeen et al., 2020*), HUMAnN 3 produces more accurate estimates of EC abundances and displays a higher species true positive rate compared to

*Figure 1 continued on next page*

*Figure 1 continued*

HUMAnN 2. In panels B–D, an asterisk ('*') indicates that the bioBakery 3 method (MetaPhlAn 3 or HUMAnN 3) scored significantly better than all other methods (repeated paired *t*-tests over synthetic metagenomes, two-tailed p<0.05).

The online version of this article includes the following figure supplement(s) for figure 1:

**Figure supplement 1.** Performance metrics (Precision, Recall, Bray-Curtis similarity) of MetaPhlAn 3, MetaPhlAn2, mOTU, and Bracken species-level profiling of the CAMI human-associated, CAMI mouse gut, and non-human datasets.

**Figure supplement 2.** (top) Scatter plots of precision, recall, and F1 score, of all the synthetic metagenomes profiled with MetaPhlAn 3 using stat_q = 0.2 (default value for MetaPhlAn 3) and stat_q = 0.1 (rho = 0.97).

**Figure supplement 3.** This figure expands *Figure 1D* from the main text to further compare HUMAnN 3, HUMAnN 2, and Carnelian on the basis of F1 score for accuracy of enzyme commission (EC) family detection, runtime (cpu-hrs), and peak memory usage (MaxRSS).

**Figure supplement 4.** Re-optimization of HUMAnN 3 based on the synphlan-humanoid metagenome and UniRef90 gold standard.

**Figure supplement 5.** This figure provides a high-resolution view of HUMAnN 3's performance in the evaluations of main-text *Figure 1D* (accuracy and performance on CAMI and non-human-associated metagenomes).

---

ChocoPhlAn 3 from the set of 16.8 k species pangenomes. The adoption of UniRef90 gene families permitted the efficient expansion of the core-gene identification procedure, which is followed by a mapping of potential core genes against all available whole microbial genomes to ensure unique marker identification (see Materials and methods). This restructuring of the marker selection process has been combined with several improvements and extensions of the algorithm, including optimized quality control during marker alignments and an estimation of the metagenome fraction composed of unknown microbes (*Supplementary file 2*).

We evaluated the taxonomic profiling performance of MetaPhlAn 3 using 118 synthetic metagenomes spanning 113 synthetic samples from the 2nd CAMI Challenge (*Fritz et al., 2019*; *Sczyrba et al., 2017*) through the OPAL benchmarking framework (*Meyer et al., 2019*). These represent typical microbiomes from five human-associated body sites and the murine gut, and we complemented them with five additional newly generated synthetic non-human-associated metagenomes (see Materials and methods). In addition to MetaPhlAn 3, the comparative evaluation considered MetaPhlAn 2.7 (*Truong et al., 2015*), mOTUs 2.51 (*Milanese et al., 2019*) (latest database available as of July 2020), and Bracken 2.5 (using a database built after the April 2019 RefSeq release) (*Lu et al., 2017*; *Wood et al., 2019*). These three profiling tools have consistently been shown to outperform other methods across multiple evaluations (*McIntyre et al., 2017*; *Meyer et al., 2019*; *Milanese et al., 2019*; *Sczyrba et al., 2017*; *Truong et al., 2015*; *Ye et al., 2019*).

MetaPhlAn 3 outperformed all the other profilers across all considered types of communities when assessing the F1 score (*Figure 1B*), which is a measure combining the fraction of species actually present in the metagenomes that are correctly detected (recall, *Figure 1—figure supplement 1*) and the fraction of species predicted to be present that were actually included in the synthetic metagenome (precision, *Figure 1—figure supplement 1*). With a very low number of false positive species detected, MetaPhlAn 3 (avg 8.51 s.d. 5.12) also maximized precision (*Figure 1—figure supplement 1*) with respect to the other tools (avg 9 s.d. 4.78 for mOTUs2 in high precision mode, the closest competitor on precision). On recall, Bracken and mOTUs2 in high-recall mode were in several cases superior to MetaPhlAn 3, but at the cost of a very high number of false positives (on average 729 species for Bracken and 39 for mOTUs2 high-recall, for a total of 86,077 and 4655 false positive species across the synthetic metagenomes). Of note, 379 of the total 1119 (33%) species in the synthetic metagenomes were not present in the database of MetaPhlAn 2, emphasizing the role of expanded isolate genome availability in the improved detection capabilities of MetaPhlAn 3. MetaPhlAn 3 can further minimize false positives by requiring a higher fraction of positive markers for positive species ('–stat_q' parameter, *Figure 1—figure supplement 2*), but overall, the F1 measure with default settings remains higher than the other evaluated tools across the panel of synthetic metagenomes in our evaluation.

In addition to more accurate species detection, MetaPhlAn 3 also quantified taxonomic abundance profiles more accurately compared to MetaPhlAn 2, mOTUs2, and Bracken based on Bray-Curtis similarity in most datasets (*Supplementary file 3*, *Figure 1C*). While it was slightly outperformed by mOTUs2 (only in high-recall mode) on the synthetic mouse gut dataset, even in this case, correlation-based measures (Pearson Correlation Coefficient between estimated and expected

relative abundances) found MetaPhlAn 3 to be more accurate (r = 0.73) than the other considered profilers (MetaPhlAn 2 r = 0.63, mOTUs2 precision r = 0.60, mOTUs2 recall r = 0.71, Bracken r = 0.43). Additionally, because Bracken estimates the fraction of reads belonging to each taxon rather than its relative abundance, we also re-normalized its estimates based on genome length of the target species. This improved Bracken's performance on taxonomic abundances (but not false positives or false negatives, see Materials and methods), but even so they were comparable with MetaPhlAn 3 in only some of the simulated environments (*Figure 1—figure supplement 1*). Overall, this confirms that MetaPhlAn 3 is superior to its previous version and is more accurate than other currently available tools in the large majority of simulated environment-specific datasets.

In addition to improvements in accuracy, MetaPhlAn 3's computational efficiency also compares favorably with alternatives and with its previous version. It is >3 x faster than MetaPhlAn 2 (10.0k vs. 2.9 k reads/s on a Xeon Gold 6140) and almost matches the speed of Bracken (11 k reads/s). Meta-PhlAn 3 memory usage is slightly higher (2.6 Gb for a complete taxonomic profiling run) than Meta-PhlAn 2 (2.1Gb), but outperforms the other methods (4.3 Gb for mOTUs2 and 32.5 Gb for Bracken, *Figure 1—figure supplement 2*, *Supplementary file 4*).

## HUMAnN 3 accurately quantifies species' contributions to community function

HUMAnN 3 functionally profiles genes, pathways, and modules from metagenomes, now using native UniRef90 annotations from ChocoPhlAn species pangenomes. We compared its performance against HUMAnN 2 (*Franzosa et al., 2018*), and the recently published Carnelian (*Nazeen et al., 2020*) when profiling the 30 CAMI and five additional synthetic metagenomes introduced above (see Materials and methods and *Figure 1*). Carnelian was selected because it was published subsequent to HUMAnN 2 and, more importantly, follows the HUMAnN strategy of estimating the relative abundance of molecular functions directly from shotgun meta-omic sequencing reads rather than assembled contigs (albeit by a different approach). While HUMAnN 2 and 3 can both natively estimate the relative abundances of a wide variety of functional features from a metagenome (by first quantifying and then manipulating UniRef90 or UniRef50 abundances), we selected level-4 enzyme commission (EC) categories as a basis for comparison with Carnelian, as the method's authors provided a precomputed index for EC quantification (*Nazeen et al., 2020*).

HUMAnN 3 produced highly accurate estimates of community-level EC abundances across the 30 CAMI metagenomes (mean ± SD of Bray-Curtis similarity = 0.93 ± 0.03, *Figure 1*). HUMAnN 2 followed with an accuracy of 0.70 ± 0.04 and Carnelian at 0.49 ± 0.04. While HUMAnN 3 benefits in part from access to a more up-to-date sequence database, we note that HUMAnN 2's database (c. 2014) predates the Carnelian method by several years, and so recency cannot be the only explanation for this trend. For example, Carnelian uses a mean sequence length per EC during abundance estimation, a choice which may contribute additional error relative to HUMAnN's sum over per-sequence estimates. We observed similar trends in accuracy among the three methods using F1 score (i.e. the harmonic mean of sensitivity and precision) to prioritize presence/absence calls over abundance (*Figure 1—figure supplement 3*). HUMAnN 3 exhibited the highest sensitivity (0.96 ± 0.05), while HUMAnN 2 and Carnelian had similar lower sensitivity scores (0.72 ± 0.05 and 0.74 ± 0.04, respectively). In contrast, HUMAnN 3 and HUMAnN 2 had similar high precision scores (0.97 ± 0.01 and 0.95 ± 0.02), while Carnelian's precision was uniquely lower (0.60 ± 0.08). This difference in precision is attributable in part to HUMAnN's use of database-sequence coverage filters to reduce false positives, an approach introduced for translated search in HUMAnN 2 and expanded to nucleotide search in HUMAnN 3 (one of a number of algorithmic refinements in HUMAnN 3 that contribute to improved accuracy and performance even when controlling for database completeness; see Materials and methods and *Figure 1—figure supplement 4*).

One of the main advantages of HUMAnN 3 (and 2) compared with other functional profiling systems (including Carnelian) is their ability to stratify community functional profiles according to contributing species. This feature is additionally more accurate and useful in HUMAnN 3 as a function of its broader pangenome catalog. Across the CAMI metagenomes, EC accuracy for species with at least 1x mean coverage depth was 0.81 ± 0.16 for HUMAnN 3 and 0.51 ± 0.15 for HUMAnN 2 (mean ± SD within-species Bray-Curtis similarity; *Figure 1*). HUMAnN 3 (via MetaPhlAn 3) additionally tended to detect more expected species in this coverage range compared with HUMAnN 2, a major driver of its improved community-level accuracy. As previously noted (*Franzosa et al., 2018*),

HUMAnN's within-species function sensitivity is naturally lower for species below 1x coverage in a sample, as many of their genes will not have been sampled at all during the sequencing process. Per-species precision, however, remained high with HUMAnN independent of coverage and, following refinements in alignment post-processing, was slightly improved in v3 compared with v2 ($0.95 \pm 0.08$ vs. $0.91 \pm 0.07$).

Carnelian was the most computationally efficient of the three methods, analyzing the CAMI metagenomes in $26.4 \pm 2.7$ CPU-hours (per-sample mean $\pm$ SD) compared with $38.1 \pm 12.8$ CPU-hours for HUMAnN 2 and $52.5 \pm 19.2$ CPU-hours for HUMAnN 3 (*Figure 1—figure supplement 3*). Trends in peak memory use (MaxRSS) were similar, with Carnelian requiring $11.9 \pm 0.0$ GB versus HUMAnN 2's $17.0 \pm 0.3$ GB and HUMAnN 3's $21.5 \pm 1.9$ GB. We attribute these differences in large part to the sizes of the sequence spaces over which the methods search: while Carnelian focuses only on a subset of sequences annotatable to EC terms, HUMAnN aims to first quantify 10 s of millions of unique UniRef90s, of which only 12.5% are ultimately annotated by ECs. The increased runtime of HUMAnN 3 compared to HUMAnN 2 is likewise attributable to the former's larger translated search database (87.3M vs. 23.9M UniRef90 sequences), as the translated search tier is the rate-limiting step of the HUMAnN algorithm even when most sample reads are explained in the preceding nucleotide-level search tiers (*Figure 1—figure supplement 5*). This phenomenon also explains the greater runtime variability of HUMAnN, as runtimes vary inversely with the (a priori unknown) fraction of sample reads explained before the translated search tier (*Franzosa et al., 2018*). Notably, by bypassing the translated search step, HUMAnN 3 could explain the majority of CAMI metagenomic reads ($70.9 \pm 9.6\%$ per sample) in only $5.8 \pm 0.8$ CPU-hours (a 9x speed-up; *Figure 1—figure supplement 5*), although this is generally only appropriate for communities known to be well-covered by related reference sequences.

Evaluations on a set of synthetic metagenomes enriched for non-human-associated species resulted in similar relative accuracy and efficiency trends among the three methods (*Figure 1* and *Figure 1—figure supplement 3*). Hence, HUMAnN 3's strong performance is not restricted to microbial communities assembled from host-associated species. Moreover, MetaPhlAn 3's improved sensitivity for non-host-associated species increased both the accuracy and performance of HUMAnN 3 relative to HUMAnN 2 (by enabling a larger fraction of reads to be explained during the faster and more accurate pangenome search step). Finally, we evaluated HUMAnN 3's accuracy at the level of individual UniRef90 protein families (*Figure 1—figure supplement 5*). As previously noted (*Franzosa et al., 2018*), the challenge of differentiating globally homologous UniRef90 protein sequences using short sequencing reads results in a reduction of community and per-species accuracy relative to broader gene families. However, because these homologs tend to share similar functional annotations, this error is smoothed out when individual UniRef90 abundances are combined in HUMAnN's downstream steps (as seen in the EC-level evaluation; *Figure 1*).

## MetaPhlAn 3 and HUMAnN 3 expand the link between the microbiome and colorectal cancer with a meta-analysis of 1262 metagenomes

To illustrate the potential of bioBakery 3's updated profiling tools and to extend our understanding of the microbial signatures in colorectal cancer (CRC), we expanded our previous work to meta-analyze both existing and newly available CRC metagenomic cohorts for a total of 1262 samples (600 control and 662 CRC samples) from nine different datasets spanning eight different countries (*Feng et al., 2015*; *Gupta et al., 2019*; *Thomas et al., 2019*; *Vogtmann et al., 2016*; *Wirbel et al., 2019*; *Yachida et al., 2019*; *Yu et al., 2017*; *Zeller et al., 2014*). The resulting integrated profiles are available for download (*Supplementary file 5*) and included in the new release of curatedMetagenomicData (*Pasolli et al., 2017*).

MetaPhlAn 3 identified a total of 1083 species detected at least once (172 considered 'prevalent' when defined as present in >5% of samples at >0.1% relative abundance), of which 505 species (52 prevalent) were previously not reported by MetaPhlAn 2 due to the expansion of the genome database (or in some cases because of changes in the NCBI taxonomy). In addition, 82 species present in the MetaPhlAn 2 database were not detected by MetaPhlAn 2 but are now identified in the samples by MetaPhlAn 3, due to the expanded sequence catalog, improved marker discovery procedure, and increased sensitivity to low-abundance species (*Thomas et al., 2019*).

We found 121 species significantly associated with CRC (FDR q < 0.05 and Q-test for heterogeneity >0.05; *Supplementary file 6*) by a meta-analysis of standardized mean differences using a

random-effects model on arcsine-square-root-transformed relative abundances (see Materials and methods). Compared to MetaPhlAn 2 when run on the same data, this includes 60 additional species that reached significance in the meta-analysis, confirming that the updated methods lead to improved biomarker discovery. Among them, three additional species not identified in previous MetaPhlAn-2-based analysis (*Thomas et al., 2019*) were among those most strongly associated with CRC (effect size >0.35): *Dialister pneumosintes*, *Ruthenibacterium lactatiformans*, and *Eisenbergiella tayi* (*Figure 2B*, *Figure 2—figure supplement 1*, *Figure 2—figure supplement 2A*). *Dialister pneumosintes* is typically oral, further reinforcing the role of oral taxa in CRC, and *R. lactatiformans* was reported as part of a consortium of bacteria able to increase colonic IFNγ+T cells (*Tanoue et al., 2019*). The increased number of species detectable by MetaPhlAn 3 also strengthened the previously observed pattern of greater richness in CRC-associated microbiomes. This pattern has been found previously both with MetaPhlAn 2 (*Thomas et al., 2019*) and with mOTUs2 (*Wirbel et al., 2019*), in large part due to low-level detection of typically oral microbes in addition to the baseline gut microbiome during CRC. Improved MetaPhlAn 3 profiling in this study allowed the difference to achieve even stronger statistical significance (*Figure 2C*).

Functional profiling of this expanded CRC meta-analysis with HUMAnN 3 identified 4.3M UniRef90 gene families, corresponding to 549 MetaCyc pathways and 2895 ECs. Out of these 4.3M gene families, the meta-analysis identified 206,296 significantly associated with CRC and controls (FDR q < 0.05), a substantial increase from the 64,315 gene families previously identified with the meta-analysis based on HUMAnN 2 profiles (FDR q < 0.05). In addition, 120 MetaCyc pathways were significantly associated with CRC (Wilcoxon rank-sum test FDR q < 0.05 and Q-test for heterogeneity >0.05) (*Figure 2—figure supplement 2B*), of which 59 (49.1%) overlapped previous results, including for example the increased abundance of starch degradation V (*Supplementary file 6*) in healthy individuals. This pathway encodes functions for extracellular breakdown of starch by an amylopullulanase enzyme, which has both pullulanase and α-amylase activity (*Flint et al., 2012*). *Bifidobacterium breve* and other *Bifidobacterium* spp have been shown to encode amylopullulanases and attach to starch particles, and they have also been reported for their potential protective role against carcinogenesis here and previously (*Sivan et al., 2015*). Among the 20 disease-associated pathways with the highest significance, only three were present in the previous meta-analysis, with the majority exhibiting significant heterogeneity in the random effects model (possibly due to the inclusion of additional geographically distinct cohorts here). Large and diverse cohorts combined with improved taxonomic and functional profiling available via bioBakery 3 thus have the possibility to extend and refine microbiome biomarkers in CRC and other conditions.

Improvements in HUMAnN 3 also allowed us to directly test specific functional hypotheses in the context of the CRC microbiome. Specifically, we previously showed that the abundance of the microbial gene encoding for the choline trimethylamine-lyase (*cutC*) responsible for production of the disease-associated trimethylamine and trimethylamine N-oxide (*Kalnins et al., 2015*; *Kummen et al., 2017*; *Oellgaard et al., 2017*; *Rath et al., 2017*) (TMA/TMAO) is significantly higher in CRC patients (*Thomas et al., 2019*). This association was previously detected using a customized ShortBRED database (*Kaminski et al., 2015*) due to the lack of appropriate reference sequences previously available to HUMAnN 2. HUMAnN 3 was instead able to directly profile relative abundances of 113 UniRef90 gene families annotated as *cutC* orthologs and identified 909 metagenomes in this data collection carrying at least one gene family annotated as *cutC*. These confirmed an increase of *cutC* relative abundance in CRC samples compared to controls (Wilcoxon rank-sum test p<0.05 in six of the nine datasets, meta-analysis p<0.0001). Interestingly, a meta-analysis performed on the relative abundances of the L-carnitine dioxygenase gene (*yeaW*), a gene also involved in trimethylamine synthesis, revealed only weak associations with disease status (Wilcoxon rank-sum test p<0.05 in three of the nine datasets, meta-analysis p=0.095, *Figure 2—figure supplement 3*, *Figure 2—figure supplement 4*), possibly reflecting a stronger effect of dietary choline on CRC risk compared to carnitine.

MetaPhlAn 3 and HUMAnN 3 also proved accurate when combining CRC microbiomes using more purely discriminative models such as random forests (RFs), reaching 0.85 average AUC for CRC (vs. control) sample classification in leave-one-dataset-out evaluations using taxonomic features (LODO, minimum 0.76 for the YachidaS_2019 and ThomasAM_2019_a datasets, maximum 0.97 for the GuptaA_2019 dataset; *Figure 2E*, *Figure 2—figure supplement 5*). As in previous studies (*Pasolli et al., 2016*; *Thomas et al., 2019*), RFs using functional features performed similarly (0.69 cross validation and 0.71 LODO ROC AUC on pathways relative abundance), indicating a tight link

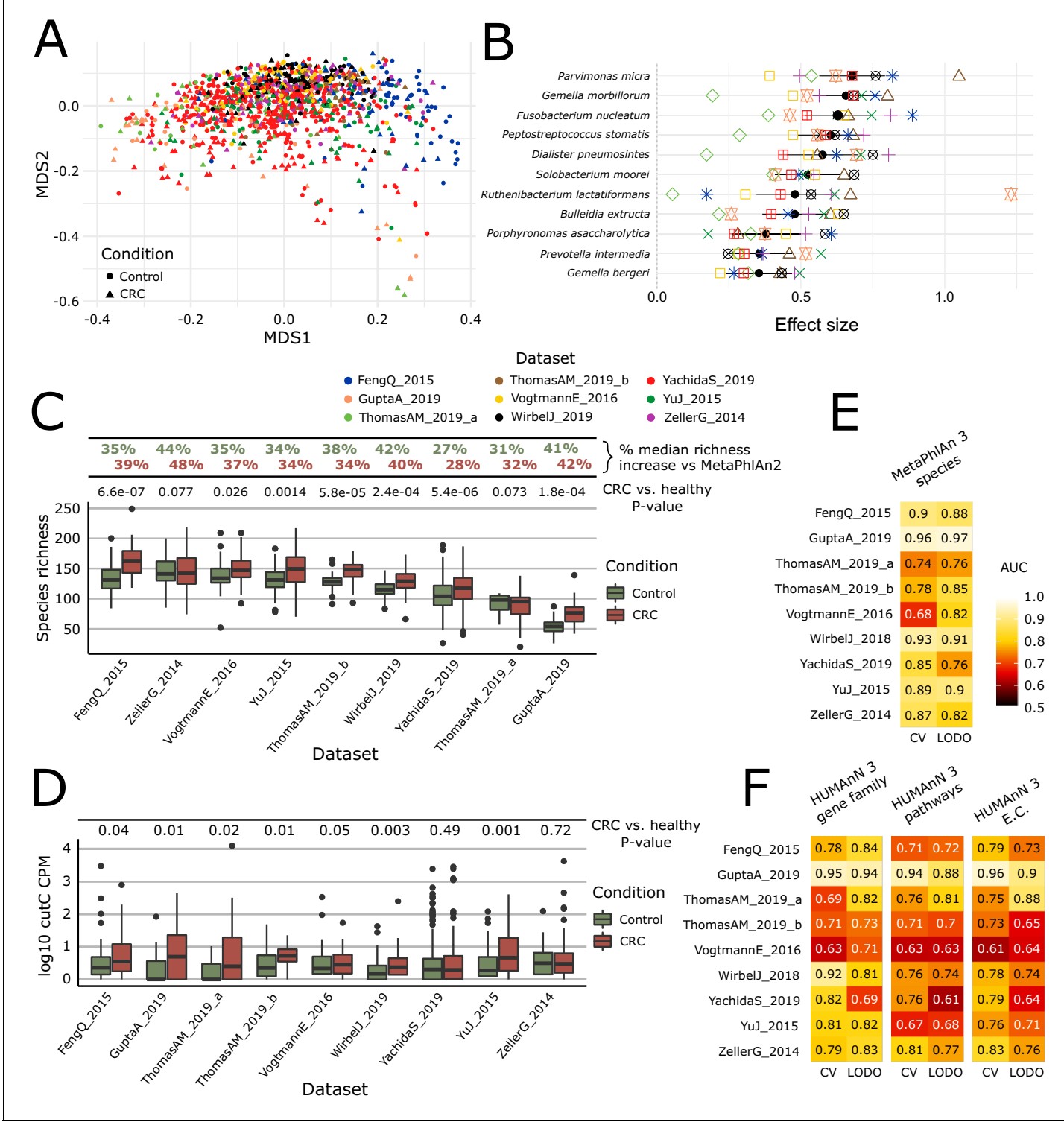

**Figure 2.** Meta-analysis with MetaPhlAn 3 and HUMAnN 3 expands taxonomic and functional associations with the CRC microbiome. (**A**) We considered a total of nine independent datasets (1262 total samples) that highly but not completely overlap (PERMANOVA p=0.001 for each single dataset when compared to all the other together; between-datasets R² = 0.14; within-dataset R² = 0.86) in composition based on ordination (multidimensional scaling) of weighted UniFrac distances (*Lozupone and Knight, 2005*) computed from the MetaPhlAn 3 species relative abundances. (**B**) Meta-analysis based on standardized mean differences and a random effects model yielded 11 MetaPhlAn 3 species significantly (Wilcoxon rank-sum test FDR p<0.05) associated with colorectal cancer at effect size >0.35 (see Materials and methods). (**C**) Species richness is significantly higher in CRC samples compared to control (Wilcoxon rank-sum test on species richness in healthy versus CRC p<0.05 in 7/9 datasets), and the expanded

*Figure 2 continued on next page*

*Figure 2 continued*

MetaPhlAn 3 species catalog detects more species compared to MetaPhlAn 2 (CRC mean median increase 37.1%, controls mean median increase 36.3%). (D) Distribution of *cutC* gene relative abundance (log10 count-per-million normalized) from HUMAnN 3 gene family profiles supporting the potential link between choline metabolism and CRC (*Thomas et al., 2019*). (E) Random forest (RF) classification using MetaPhlAn 3 features and HUMAnN 3 features (F) confirms that CRC patients can be predicted at (treatment-naive) baseline from the composition of their gut microbiome with performances reaching ~0.85 in cross-validated (CV) or leave-one-dataset-out (LODO) ROC AUC (see Materials and methods).

The online version of this article includes the following figure supplement(s) for figure 2:

**Figure supplement 1.** Log-transformed relative abundances of the top 20 MetaPhlAn 3 species associated with colorectal cancer (A) and top 10 most abundant species (B) identified with a meta-analysis on 1262 samples.

**Figure supplement 2.** Meta-analysis of the CRC datasets on the MetaPhlAn 3 species-level relative abundances (A) and relative abundance of MetaCyc pathway profiles generated with HUMAnN 3 (B).

**Figure supplement 3.** Forest plot reporting effect sizes calculated using a meta-analysis of standardized mean differences and a random effects model on *cutC* (A) and *yeaW* (B) relative abundances between CRC and control samples.

**Figure supplement 4.** Distribution of *yeaW* gene relative abundance (log10 count-per-million normalized) extracted from HUMAnN gene family profiles.

**Figure supplement 5.** Features identified by the random-forest analysis on the species profiled with MetaPhlAn 2 and MetaPhlAn 3 using different values of q_stat, and by HUMANn 3 grouping UniRef90 in MetaCyc pathways and Enzyme Commission numbers.

between strain-specific taxonomy and gene carriage in this setting. When the classification model was used for assessing features' importance, several new taxa were identified compared to Meta-PhlAn 2 and metabolic pathways or EC-numbers relative to HUMAnN 2 (*Figure 2—figure supplement 5*), further confirming the relevance of the new reference sequences and annotations available to be profiled in bioBakery 3.

## Longitudinal taxonomic and functional meta-omics of IBD

To further demonstrate the utility of MetaPhlAn 3 and HUMAnN 3 on combined meta-omic sequencing datasets, including identification of expression-level biomarkers, we applied the updated methods to 1635 shotgun metagenomes (MGX) and 817 shotgun metatranscriptomes (MTX) derived from the stool samples of the HMP2 Inflammatory Bowel Disease Multi-omics Database (IBDMDB) cohort (http://ibdmdb.org; see Materials and methods). Compared with previously published profiles of the samples generated with MetaPhlAn 2 and HUMAnN 2 (*IBDMDB Investigators et al., 2019*; *Figure 3A*), the v3 methods' profiles (i) identified more species pangenomes (MGX medians 40 vs. 48, MTX medians 40 vs. 47); (ii) explained larger fractions of sample reads by mapping to pangenomes (MGX medians 54 vs. 63%, MTX medians 12 vs. 22%); and (iii) explained larger total fractions of sample reads after falling back to translated search (MGX medians 69 vs. 75%, MTX medians 20 vs. 31%). Note that reduced MTX mapping rates (relative to MGX rates) result from enrichment for high-quality but non-coding RNA reads, which are unmapped by design in both HUMAnN 2 and 3. The v3 profiles thus promise increased understanding even of an already well-characterized dataset.

To that end, we applied a mixed-effects model to identify microbial biomarkers of disease activity within the Crohn's disease (CD) and ulcerative colitis (UC) subpopulations of the HMP2 cohort (see Materials and methods). More specifically, we examined HUMAnN 3-based abundance profiles of EC families from 817 paired HMP2 metagenomes and metatranscriptomes in search of differences in functional activity between active (dysbiotic) and inactive (non-dysbiotic) time points from longitudinally sampled CD and UC patients. We identified 558 ECs whose residual expression was significantly different (FDR $q < 0.05$) in active CD compared with inactive CD (*Figure 3B*): a 66% increase compared to an identical analysis incorporating EC abundance profiles generated by HUMAnN 2 (*Figure 3—figure supplement 1*). We identified only one EC that was differentially expressed in active UC (protein O-GlcNAcase, EC 3.2.1.169). This relative absence of biomarkers for active UC may result both from its generally more benign phenotype (*IBDMDB Investigators et al., 2019*) and from the smaller number of active UC samples (n = 23) compared with active CD samples (n = 76); as a result, we focused our subsequent analyses on expression differences within the CD subcohort.

Of the >500 significantly differentially expressed ECs in active CD, all but one were 'over-expressed' (i.e. their residual expression after controlling for DNA copy number was higher than expected in active CD; see *Figure 3B*). Hence, while many species (and their encoded functions) are known to be lost entirely during active IBD (*IBDMDB Investigators et al., 2019*), it seems to be rare

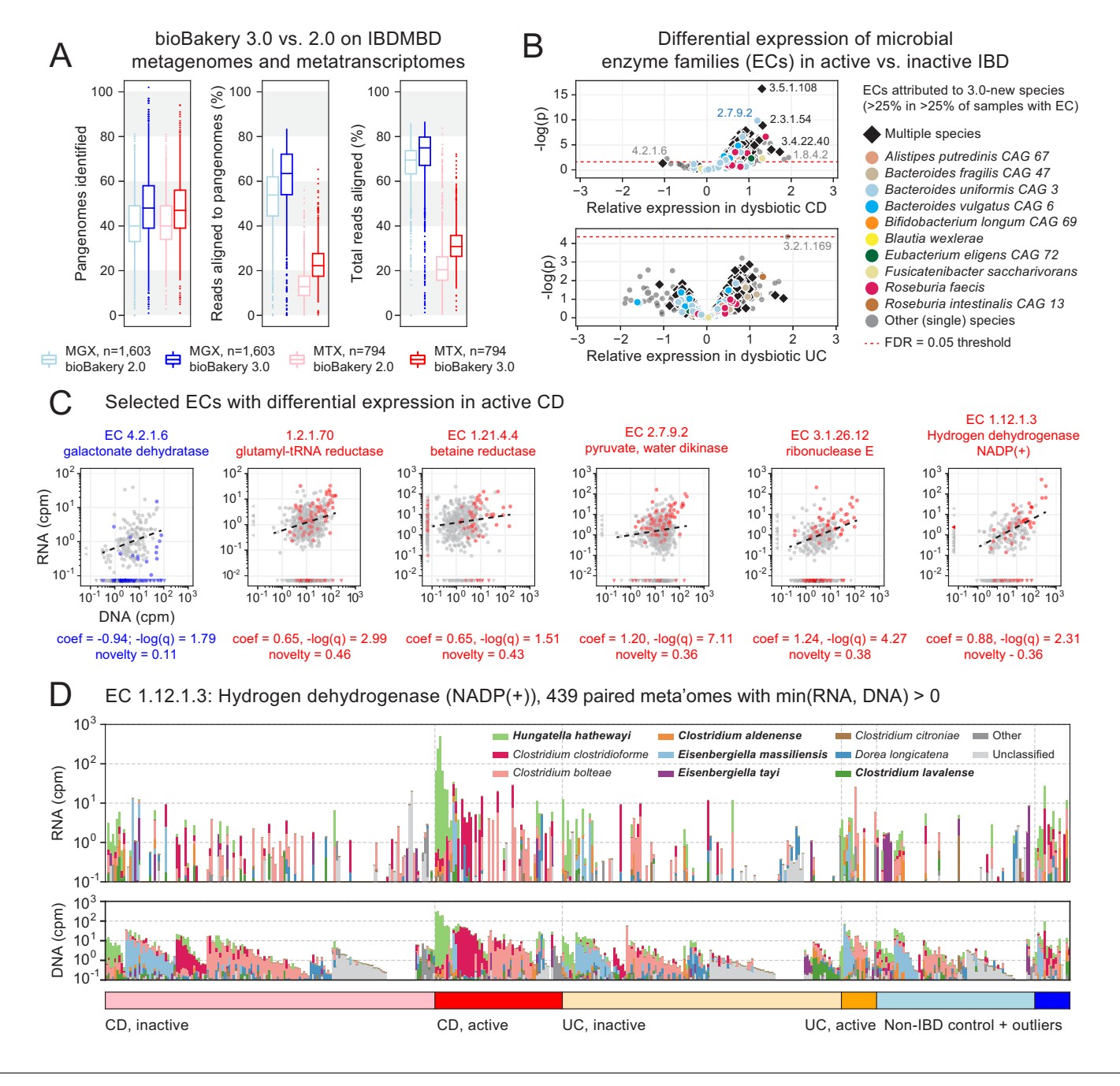

**Figure 3.** Longitudinal taxonomic and functional meta-omics of IBD. (A) Comparison of MetaPhlAn and HUMAnN profiles of IBDMDB metagenomes and metatranscriptomes using v2 and v3 software (sequencing data and v2 profiles downloaded from http://ibdmdb.org). (B) > 500 Enzyme Commission (EC) families were significantly [linear mixed-effects (LME) models, FDR q < 0.05] differentially expressed in active CD relative to inactive CD; only a single EC met this threshold for active UC. ECs (points) are colored to highlight large contributions from one or more species that were new or newly classified in MetaPhlAn 3 (independent of the strength of their association with active IBD). (C) Selected examples of EC families that were differentially expressed in active CD. Colored points correspond to active CD samples; all other samples are gray. The first example (blue) is the only EC to be down-regulated in active CD (as indicated by CD active samples falling below the best-fit RNA vs. DNA line). To match the associated LME models (see Materials and methods), best-fit lines exclude samples where an EC's RNA or DNA abundance was zero (such samples are shown as triangles in the x:y margins). (D) Species contributions to RNA (top) and DNA (bottom) abundance of EC 1.12.1.3. The seven strongest contributing species are colored individually; bold names indicate new species in MetaPhlAn 3. Samples are sorted according to the most abundant contributor and then grouped by diagnosis. The tops of the stacked bars (representing community total abundance) follow the logarithmic scale of the y-axis; species' contributions are linearly scaled within that height.

*Figure 3 continued on next page*

*Figure 3 continued*

The online version of this article includes the following figure supplement(s) for figure 3:

**Figure supplement 1.** We repeated the differential expression analysis described in the main text using metagenomic and metatranscriptomic profiles of enzyme (EC) abundances from the IBDMDB population based on HUMAnN 2.11.0.

for functions to be maintained by the community but not utilized. The one notable example of an 'under-expressed' function was galactonate dehydratase (EC 4.2.1.6; *Figure 3C*). This enzyme was encoded and highly expressed by *Faecalibacterium prausnitzii* in both control and inactive CD samples. While galactonate dehydratase was still metagenomically abundant in active CD (where it was contributed primarily by *Escherichia coli*), it was not highly expressed under those conditions. Related observations were made previously using a mouse model of colitis monocolonized with commensal *E. coli* (*Patwa et al., 2011*). There, microarray-based measurements found a number of enzymes in the galactonate utilization pathway, including galactonate dehydratase, to be among the most strongly down-regulated in comparison with wild-type mice. These results suggest that galactonate metabolism is either infeasible (e.g. due to low bioavailability) or otherwise suboptimal (e.g. due to the presence of preferred energy sources) in the inflamed gut, thus leading to its down-regulation by 'generalist' pathobionts like *E. coli*.

From the many over-expressed functions in active CD, we focused for illustrative purposes on examples that were encoded non-trivially by species either new or newly classified in MetaPhlAn 3 ('3.0-new species'; *Figure 3C*). To aid in this process, we defined an *h*-index-inspired 'novelty' score ($s$) for each EC equal to the largest percentile $p$ of samples with the EC in which $p$ percent of its copies were contributed by 3.0-new species. For example, an EC with $s = 0.25$ indicates that at least 25% of the EC's copies were from 3.0-new species in at least 25% of samples with the EC. The previously mentioned galactonate dehydratase thus had a low novelty score ($s = 0.11$) resulting from dominant contributions of *F. prausnitzii* and *E. coli* (which are not new to MetaPhlAn 3).

Conversely, the highest novelty score was observed for glutamyl-tRNA reductase (EC 1.2.1.70, $s = 0.46$), a highly-transcribed housekeeping gene that received large contributions from the 3.0-new species *Roseburia faecis*, *Phascolarctobacterium faecium*, and *Ruminococcus bicirculans*. Betaine reductase (EC 1.21.4.4, $s = 0.43$), instead, is much more specific and was contributed in part by 3.0-new species *Hungatella hathewayi*; this is notable as a rare example of a function that was often detectable from community RNA but not DNA (indicating high expression from a small pool of gene copies). Pyruvate, water dikinase (EC: 2.7.9.2) and Ribonuclease E (EC 3.1.26.12) were among the strongest signals of over-expression in active CD by both effect size and statistical significance; these functions were also characterized by large contributions of 3.0-new species ($s = 0.36$ and $0.38$, respectively). Ribonuclease E and a final example, hydrogen dehydrogenase NADP(+) (EC 1.12.1.3), are also representative of the degree to which metagenomic copy number (DNA abundance) tends to be a strong driver of transcription (RNA abundance) in the gut microbiome, and thus the need to account for the former when estimating functional activity. The 3.0-new *H. hathewayi* expresses this enzyme highly in a subset of active CD samples, thus contributing to the enzyme's overall association with active CD.

## Population-scale subspecies genetics (StrainPhlAn) and pangenomics (PanPhlAn) of *Ruminococcus bromii*

Strain-level characterization of taxa directly from metagenomes is an effective cultivation-free means to profile the population structure of a microbial species across geography or other conditions (*Scholz et al., 2016*; *Truong et al., 2017*) and to track strain transmission (*Ferretti et al., 2018*). These functionalities are incorporated into (i) StrainPhlAn 3, which infers strain-level genotypes by reconstructing sample-specific consensus sequences from MetaPhlAn 3 markers (*Zolfo et al., 2019*) (ii) PanPhlAn 3, which identifies strain-specific combination of genes from species' pangenomes; and (iii) PhyloPhlAn 3, which performs precise phylogenetic placement of isolate and metagenome-assembled genomes (MAGs) using global and species-specific core genes (*Asnicar et al., 2020*) (see Materials and methods). ChocoPhlAn 3 automatically quantifies and annotates the distinct types of conservation metrics necessary to identify these markers, all updated in bioBakery 3 (*Supplementary file 2*).

*Ruminococcus bromii* is a common gut microbe that is surprisingly understudied due to its fastidious anaerobicity and general non-pathogenicity (*Ze et al., 2012*). *R. bromii* is prevalent in over half of typical gut microbiomes, but large-scale comparative genomic analyses of this species are not available with only one previous investigation (*Mukhopadhya et al., 2018*) limited to the five reference genomes available at the time. This made *R. bromii* population genetics, geographic association, and genomic variability of particular interest to assess via StrainPhlAn and PanPhlAn. From the meta-analysis of 7783 gut metagenomes integrated for a previous study (*Pasolli et al., 2019*), we considered the 4077 metagenomes in which *R. bromii* was found present with a relative abundance above 0.05% according to MetaPhlAn 3. StrainPhlAn analysis based on single-nucleotide variants (SNVs) of the 124 *R. bromii*-specific marker genes across the 3375 samples with sufficient markers' coverage (see Materials and methods) revealed a complex population structure not previously recapitulated by the only 15 genomes available from isolate sequencing (*Figure 4A*). Sub-clade prediction (see Materials and methods) highlighted two sub-species clades that are particularly divergent within the phylogeny (*Figure 4—figure supplement 1C–D*); interestingly, the first one (Cluster 1) is mainly composed of strains retrieved from Chinese subjects and from cohorts with a rural and a more pre-industrial traditional lifestyle and diet (*Pasolli et al., 2019*; *Figure 4A*; Cluster 1; Fisher's exact test p<2.2e-16). StrainPhlAn 3 can thus rapidly reconstruct complex strain-level phylogenies from metagenomes (5,700 s using 20 CPUs), and with the integration of PhyloPhlAn 3's improvements specifically for strain-level manipulation of alignments and phylogenies (*Asnicar et al., 2020*), surpasses the previous version of the software in accuracy and sensitivity (67.4% more strain profiled, *Figure 4—figure supplement 1A–B*).

StrainPhlAn 3 also extends the ability of reference-based approaches to infer the genetic identity of strains across samples as previously explored (*Ferretti et al., 2018*; *Truong et al., 2017*). Specifically for *R. bromii*, different individuals tend to carry different strains diverged with a roughly normal distribution of genetic identities (mean 3.54e-3 normalized phylogenetic distance, *Figure 4B*). However, the genetic differences between Cluster 1 and Cluster 2 were generally greater, with a lower peak and higher distances (mean 6.1e-3, *Figure 4B*). For carriers of either clade, within-subject strain retention tended to be high as expected (i.e. low divergence); at distinct time points (average 261.35 s.d. 239.86 days, first quartile 72 days, third quartile 386 days, 3537 comparisons in total), most of the strain distances (76.4%) approached zero (compared to 1% of comparisons for inter-individual differences, *Figure 4B*). In addition to detecting these two genetically distinct clades and quantifying within-individual strain retention, a final distribution of higher intra-individual distances clearly captured (rare) strain replacement by *R. bromii* strains (i) in the same or (ii) in a different main cluster in the species' phylogeny. Mother-infant pairs showed a similar dynamic (*Figure 4B*), with sporadic vertical transmission (~33.3%) (*Ferretti et al., 2018*; *Korpela et al., 2018*; *Yassour et al., 2018*) mixed with strain loss, replacement, and acquisition from other environmental or human sources (*Korpela et al., 2018*). This analysis highlighted the high precision of StrainPhlAn 3 in detecting strain identity across samples and thus the potential of using it for tracking the transmission network of specific individual strains within and between subjects.

PanPhlAn 3 provides a complementary form of strain analysis by constructing pangenome presence-absence (rather than individual nucleotide variant) genotypes (see Materials and methods). Using eight *R. bromii* reference genomes, PanPhlAn 3 revealed the presence of 6151 UniRef90 pangenes across 2679 samples with sufficient depth to permit confident strain-specific gene repertoire reconstruction (*Figure 4C*). This mirrored the genetic divergence of *R. bromii* Clusters 1 and 2, while also highlighting a range of functional differences annotatable to genes unique to the two clusters: Cluster 1 and Cluster 2 showed a total of 797 and 601 UniRef90 families specific to them (Fisher's exact test, FDR q < 0.05). Although most of these gene families do not have precise functional annotations, these sets of genes should be prioritized in experimental characterization efforts to unravel the sub-species diversity of *R. bromii*, and UnireRef90-to-GO ID mapping also highlighted an enrichment of membrane proteins in Cluster 2. Interestingly, other clusters of co-occurring genes were independent of phylogenetic structure and also verified to be on the same locus on at least two reference genomes in the PanPhlAn 3 database (*Figure 4C*) providing a new approach at identifying and annotating potential laterally-mobile elements.

StrainPhlan 3 and PanPhlAn 3 can thus be combined with PhyloPhlAn 3 (*Asnicar et al., 2020*) and HUMAnN 3 to provide multiple, complementary, culture-independent means to investigate the strain-level diversity of taxa in the microbiome, from new data or by re-using thousands of publicly

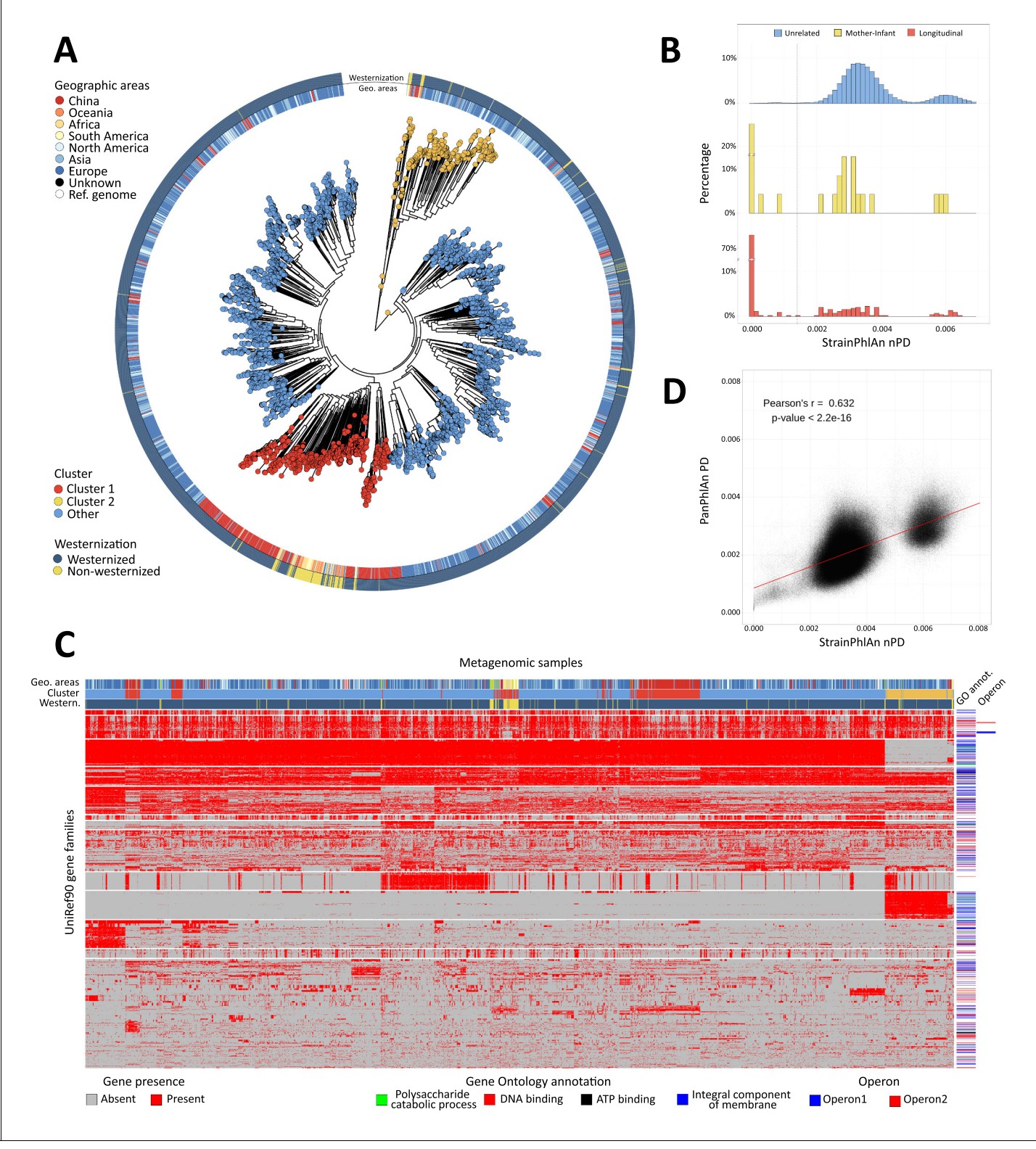

**Figure 4.** Population-scale strain-level phylogenetic and pangenomic analyses of *Ruminococcus bromii* from over 4000 human gut metagenomes. (**A**) StrainPhlAn 3 profiling revealed stratification of *Ruminococcus bromii* clades with genetic content and variants frequently structured with respect to geographic origin and lifestyle. Genetically divergent subclades were identified, labeled as 'Cluster 1' (mainly composed of strains retrieved from Chinese subjects, Fisher's exact test p<2.2e-16) and a subspecies-like Cluster 2. (**B**) Strain tracking of *R. bromii*. While unrelated individuals from diverse populations very rarely share highly genetically similar strains, pairs of related strains are readily detected by StrainPhlAn from longitudinal samples

*Figure 4 continued on next page*

*Figure 4 continued*

from the same individuals (quantifying short- and medium-term strain retention at about 75%) and in mother-infant pairs (confirming this species is at least partially vertically transmitted). Normalized phylogenetic distances (nPD) were computed on the StrainPhlAn tree. (p<0.003 two-sample Kolmogorov–Smirnov test on all the three distributions) (C) PanPhlAn 3 gene profiles of *R. bromii* strains from metagenomes highlights the variability and the structure of the accessory genes across datasets (core genes were removed for clarity). A total of 6151 UniRef90 gene families from the *R. bromii* pangenome were detected across the 2679 of the 4077 samples in which a strain of this species was present at a sufficient abundance to be profiled by PanPhlAn. The 13 highest-rooted gene clusters are shown, highlighting co-occurrence of blocks likely to be functionally related. The most common GO annotations are also reported together with two operons containing genes verified to be on the same locus by analysis of the reference genomes in the PanPhlAn 3 database. (D) Genetic (SNV on marker genes from StrainPhlAn 3) and genomic (gene presence/absence from PanPhlAn 3) distances between *R. bromii* strains are correlated (Pearson's r = 0.632, p-value<2.2e-16) pointing at generally consistent functional divergence in this species.

The online version of this article includes the following figure supplement(s) for figure 4:

**Figure supplement 1.** Comparison between StrainPhlAn (**A**) and StrainPhlAn 3 (**B**) strain level profiling capabilities.

available metagenomes. It is notable that these approaches tend to be consistent with each other (e.g. for *R. bromii,* Pearson's r = 0.632, p<2.2e-16, *Figure 4D*), while providing different benefits and drawbacks: PanPhlAn used with HUMAnN input is computationally efficient, used from whole pangenomes has higher sensitivity, and StrainPhlAn tends to have higher specificity. As these methods unravelled the population genomic structure of *R. bromii* that was not previously known, they can be similarly applied to hundreds of other host-associated or environmental microbial species to uncover their phylogenetic, functional, and transmission characteristics. Together, the bioBakery 3 components provide an integrated platform for applying strain-level comparative genomics, taxonomic, and functional profiling to meta-omic microbial community studies.

## Discussion

Here, we introduce and validate the set of expanded microbial community profiling methods making up the bioBakery 3 platform, including quality control (KneadData), taxonomic profiling (MetaPhlAn), strain profiling (StrainPhlAn and PanPhlAn), functional profiling (HUMAnN), and phylogenetics (PhyloPhlAn), largely relying on the underlying data resource of ChocoPhlAn 3 genomes and pangenomes. These modules are each more accurate and, often, more efficient than their previous versions and current alternatives, particularly for challenging (e.g. non-human-associated) metagenomes and for multi-omics (e.g. metatranscriptomes). While the improvements were in large part a consequence of the much larger database of reference genomes that the system can now handle, additional algorithmic changes (*Supplementary file 2*) were instrumental to provide more complete reporting and higher accuracy for references already available in previous releases. In the process of these evaluations, we detected three species newly associated with CRC (*Dialister pneumosintes, Ruthenibacterium lactatiformans,* and *Eisenbergiella tayi*), over 500 enzyme families metatranscriptomically upregulated by diverse microbes in IBD, and two new phylogenetically, genomically, and biogeographically distinct subclades of *Ruminococcus bromii*.

These results highlight the degree to which meta-omic approaches can now realize the potential of culture-independent sequencing for characterizing microbial community dynamics, interactions, and evolution that are only active in situ and not in vitro. Since early studies of environmental and host-associated microbial communities (*Gill et al., 2006*; *Tyson et al., 2004*; *Venter et al., 2004*), it has been clear that many aspects of intercellular and inter-species signaling, short- and long-term evolution, and regulatory programs are exercised by microbes in their natural settings and extremely difficult to recapitulate in a controlled setting. This is supported by the extent to which 'dark matter' not previously characterized in the laboratory pervades host-associated and (especially) environmental metagenomes (*Parks et al., 2017*), with most communities containing a plurality, majority, or sometimes supermajority of novel and/or uncharacterized sequences (*Almeida et al., 2019*; *Pasolli et al., 2019*). bioBakery 3 begins to overcome this challenge by combining a greatly expanded set of reference sequences with ways of 'falling back' gracefully when encountering new sequences, while also paving the way for further integration of assembly-based discovery in the future (discussed below). Critically, this now permits large collections of meta-omes to be used in ways only previously possible with large isolate genome or transcriptome collections, for example

strain-level integrative comparative genomics, near-real-time epidemiology and evolution, and detailed gene content prediction and metabolic modeling. Results such as the heterogeneity of maternal-infant strain transmission and retention, or the globally stratified distribution of subspecies clades, would be extremely challenging to discover by other means.

Methodologically, it is notable that these new meta-omic analysis types have been enabled by several years of improved experimental fidelity, denoising, and quality control approaches. These effectively retain only the 'best' subset of reads from large, noisy meta-omes for each analysis of interest, for example only the most unique sequences for taxonomic identification, or only the most evolutionarily informative loci for phylogeny. Meta-omes are uniquely positioned for broad reuse and discovery since different 'best' subsets of each dataset can be used to answer different questions. The development of meta-omic analysis methods thus parallels that of genome-wide association studies or transcriptomics, in as much as early methods were later refined to provide much greater accuracy and scalability through removal of low-quality measurements, within- and between-study normalization approaches, statistical methods to reliably separate signal from noise, and biological annotation of previously uncharacterized loci. Similarly, methods for amplicon-based community profiling have progressed from noise- and chimera-prone stitching and clustering to near-exact sequence variant tracking (*Callahan et al., 2016*). Fortunately, continued decreases in sequencing prices and increases in protocol efficiency have now made shotgun meta-omics nearly as affordable as amplicon sequencing in many settings. The challenge, of course, is that each metagenome combines many different noise sources: there is no single, whole genome to finish; host, microbial, and contaminant sequences are not always easily differentiated; there is no one set of 'true' underlying variants (since each organism might be represented by multiple strains); and millions of microbial gene products remain functionally uncharacterized (*Thomas and Segata, 2019*).

Notably, bioBakery provides one of very few environments currently capable of integrating both metagenomes and metatranscriptomes to begin overcoming these uncertainties (*Franzosa et al., 2018*). As introduced above, microbial community transcriptomes can be highly unintuitive to interpret, as transcript abundance is always influenced both by expression level and by underlying DNA copy number, that is abundance of the expressing taxon. Since both sequence-based DNA and RNA profiles are typically compositional (relative, not absolute, abundances), there is not always a simple way to account for these effects. HUMAnN 3 provides initial within- and between-species normalization options that can be combined with the statistical models of differential expression described here, making for example the >500 transcripts overexpressed in Crohn's disease particularly noteworthy. *Hungatella hathewayi* was uniquely responsible for many of these, an organism not previously associated with IBD in humans (*Schaubeck et al., 2016*). While many of its overexpressed transcripts are core or housekeeping processes, indicative of general bioactivity in the inflamed gut (comparable to that of for example *Escherichia coli IBDMDB Investigators et al., 2019*), others such as betaine reductase are much more specific. This enzyme contributes directly to trimethylamine (TMA) formation (*Rath et al., 2019*), one of the more noteworthy microbial metabolites implicated in human disease via its transformation to proatherogenic trimethylamine-oxide (TMAO) (*Tang et al., 2013*). Conversely, the only transcript differentially regulated in ulcerative colitis, underexpressed *F. prausnitzii* galactonate dehydratase, contrasts its utility in polysaccharide degradation under non-inflamed conditions with the upregulation of alternative, more host-antagonistic energy sources in *E. coli* during inflammation (*IBDMDB Investigators et al., 2019*). Both of these examples are only analyzable due to the highly specific assignment of meta-omic reads to individual community members' gene families, in combination with appropriate downstream statistical methods for multi-omics.

Finally, it is striking that metagenomically-derived comparative genomics has only recently been able to reach the scale and scope previously possible with microbial isolates. The genomic epidemiology of pathogens has driven the latter – recently in viral outbreaks such as COVID-19 (*Lu et al., 2020*) and Ebola (*Gire et al., 2014*), and in many bacterial conditions such as cholera (*Weill et al., 2017*) or pneumonia (*Croucher et al., 2011*). Since metagenomes can simultaneously access all community members with relatively little bias, such studies are now possible with organisms previously overlooked due to the absence of obviously associated phenotypes or convenient culture techniques (*Manara et al., 2019*; *Pasolli et al., 2019*). *Ruminococcus bromii* is one such example; despite being over 50% prevalent among typical human gut communities, only 15 isolates were previously sequenced, precluding epidemiology or phylogenetics. In addition to making a novel sub-species

phylogenetic and biogeographic structure apparent, the combination of MetaPhlAn, HUMAnN, Pan-PhlAn, StrainPhlAn, and PhyloPhlAn together confirmed that most *R. bromii* strains are 'personal' (i.e. specific to and retained within individuals, like most microbiome members), rarely transmissible across hosts, and that genomic differences characterize each subspecies (suggesting a degree of functional adaptation and specialization). Such results are in principle possible with any combination of metagenomic and isolate taxa and genes of interest, richly integrating culture-independent data with hundreds or thousands of isolate genomes.

Of course, many challenges remain both for improvement of the bioBakery platform and for the field as a whole. Both experimental and computational accessibility of non-bacterial microbial community members remains limited. While accurate, bioBakery 3's capacity for non-bacterial profiling is only slightly improved from the previous version by the expansion of available eukaryotic microbial reference sequences. These components of metagenomes – and, for RNA viruses, metatranscriptomes – are often measured with surprising heterogeneity during the initial generation of sequencing data themselves (*Zolfo et al., 2019*), suggesting necessary improvements in analytical quality control and normalization as well. The visibility of species with particularly high genetic diversity within individual communities also remains limited; in most cases, only the most dominant strain of each taxon per community is currently analyzable, again for both experimental (e.g. sequencing depth) and analytical reasons (*Quince et al., 2017*). This is true both for reference-based and for assembly-based approaches, the latter of which are often also stymied by highly diverse taxa (*Pasolli et al., 2019*). A final area of improvement for bioBakery, relatedly, is the increased integration between reference-based and assembly-based approaches – begun here via PhyloPhlAn 3 – in order to better leverage MAGs (*Almeida et al., 2021*), SGBs (*Pasolli et al., 2019*), and novel gene families.

We thus anticipate improved integration of reference- and assembly-based meta-omic analyses to be one of the main areas of future development for bioBakery, along with expanded methods for other types of multi-omics in addition to transcription. There will also be a continued focus on quality control and precision, enabling new types of functional analysis within microbial communities (e.g. bioactivity and gene function prediction) without sacrificing sensitivity to rare or novel community members. We also expect to better-integrate long-read sequencing data, both via long-read mapping to reference sequences and through their contributions to better assembly-based approaches. Finally, we are also committed to the platform's availability with well-documented, open-source implementations, training material, and pre-built locally-executable and cloud-deployable packaging. Feedback on any aspect of the methods or their applications in diverse host-associated or environmental microbiome settings can be submitted at https://forum.biobakery.org, and we hope bioBakery will continue to provide a flexible, convenient, reproducible, and accurate discovery platform for microbial community biology.

## Materials and methods

bioBakery 3 is a set of computational methods for the analysis of microbial communities from meta-omic data that produce taxonomic, functional, phylogenetic, and strain-level profiles to be interpreted directly or included in downstream statistical analyses (*Figure 1A*). After read-level quality control by KneadData, MetaPhlAn 3 estimates the set of microbial species (and corresponding higher taxonomic clades) present in a sample and their relative abundances. StrainPhlAn 3 deepens genetic characterization by refining strain-level genotypes of species identified by MetaPhlAn 3. HUMAnN 3 focuses instead on the identification and quantification of the molecular functions encoded in the metagenome or expressed in the metatranscriptome, which can be resolved by PanPhlAn 3 into gene presence-absence strain-level genotypes. PhyloPhlAn 3, as previously reported (*Asnicar et al., 2020*), provides a comprehensive means to interpret the draft genomes produced by assembly-based metagenomic tools. These bioBakery 3 modules are generally based on an underlying dataset of functionally-annotated isolate microbial genes and genomes produced by ChocoPhlAn 3 to quality-control and annotated UniProt derivatives. This currently includes 99,227 genomes and 87.3M gene families, almost 100-fold greater than the data types included in the first bioBakery release (*Segata and Huttenhower, 2011*).

## The AnADAMA scientific workflow manager

Most bioBakery 3 tools are integrated into reproducible workflows (the 'bioBakery workflows', http://huttenhower.sph.harvard.edu/biobakery_workflows) using the AnADAMA (Another Automated Data Analysis Management Application) task manager, currently v2 (http://huttenhower.sph.harvard.edu/anadama2). Briefly, this wraps doit (http://pydoit.org), a Python-based dependency manager, to provide a simple but scalable language for analysis task definition, version and provenance tracking, change management, documentation, grid and cloud deployment of large compute tasks, and automated reporting. AnADAMA operates in a make-like manner using targets and dependencies of each task to allow for parallelization. In cases where a workflow is modified or input files change, only those tasks impacted by the changes will be rerun. Essential information from all tasks is recorded, using the default logger and command line reporters, to ensure reproducibility. The information logged includes command line options provided to the workflow, the function or command executed for each data modification task, versions of tracked executables, and any output and data products from each task. It can optionally be used to chain together subsequent bioBakery 3 tasks and/or to parallelize them efficiently across multiple files or datasets.

## KneadData read-level quality control

bioBakery 3 includes a simple quality control module for raw sequences, KneadData (http://huttenhower.sph.harvard.edu/kneaddata), which automates a set of typical best practices for raw metagenome and metatranscriptome read cleaning and validation. These include:

- Trimming of (1) low-quality bases (default: 4-mer windows with mean Phred quality <20), (2) truncated reads (default:<50% of pre-trimmed length), and (3) adapter and barcode contaminants using Trimmomatic (*Bolger et al., 2014*).
- Removal of overrepresented sequences (default:>0.1% frequency) using FastQC (*Andrews S, 2010*) and low-complexity sequences using TRF (*Benson, 1999*).
- Depletion of host-derived sequences by mapping with bowtie2 (*Langmead and Salzberg, 2012*) against an expanded human reference genome (including known 'decoy' and contaminant sequences *Breitwieser et al., 2019*) and optionally other hosts (e.g. mouse) reference genomes and/or transcriptomes.
- Depletion of microbial ribosomal and structural RNAs by mapping against SILVA (*Yilmaz et al., 2014*) in metatranscriptomes.

It is recommended that KneadData be applied to raw sequences prior to further analyses, and the bioBakery workflows do this for all sequence types by default.

## The ChocoPhlAn 3 pipeline

We developed the ChocoPhlAn pipeline to organize microbial reference genomes according to their taxonomy and to compute the relevant sequence and annotation data for subsequent bioBakery modules. At a high level, after retrieval of UniProt genomes and gene annotations, species-specific pangenomes (i.e. the set of gene families of a species present in at least one of its genomes) are generated using all the microbial reference genomes passing initial quality control. Core genomes (i.e. gene families present in all the genomes of a species) are then identified from the whole set of pangenomes and used as markers in PhyloPhlAn 3. Core genomes are also processed for the extraction of unique marker genes (i.e. core gene families uniquely associated with one species) that constitute the marker database for MetaPhlAn 3 and StrainPhlAn 3. Finally, functionally annotated pangenomes are processed to serve as references for PanPhlAn 3 and HUMAnN 3.

### Data retrieval

ChocoPhlAn relies on the UniProt core data resources (*The UniProt Consortium, 2019*) (release January 2019) and on the NCBI taxonomy and genomes repositories (*NCBI Resource Coordinators and Coordinators, 2014*) (release January 2019). The two basic sequence data types considered in ChocoPhlAn are the raw genomes of all available microbes and all the microbial proteins/genes identified on these genomes. The main supporting structure for a genome is the underlying microbial taxonomy, whereas the microbial proteins are organized in protein families clustered at multiple stringency parameters.

We adopted the NCBI taxonomy database (*NCBI Resource Coordinators and Coordinators, 2014*) for use by ChocoPhlAn as it is the one on which our genomic repository, UniProt, is also based. The full taxonomy was downloaded from the NCBI FTP server (ftp.ncbi.nlm.nih.gov/pub/taxonomy/) on January 24 2019. We identified and tagged species with 'unidentified', 'sp.", 'Candidatus', "bacterium ", and several other keywords as low-quality species. Specifically, the regular expressions used to filter low-quality taxonomic annotations are:

"(C|c)andidat(e|us) | _sp(_.*|$) | (.*_|)(b|B)acterium(_.*|) |. *(eury|)archaeo(n_|te|n$).* |. *(endo|) symbiont.* |. *genomosp_.* |. *unidentified.* |. *_bacteria_.* |. *_taxon_.* |. *_et_al_.* |. *_and_. * |. *(cyano|proteo|actinobacterium_.*)"

All reference genomes available through UniProt Proteomes and linked to the public DDBJ, ENA, and GenBank repositories were then considered. Genomes are included by UniProt into UniProt Proteomes only if they are fully annotated and have a number of predicted CDSs falling within a statistically defined range of published proteomes from neighbouring species (*What are proteomes, 2020*). We considered all UniProt Proteomes genomes assigned to the archaeal and bacterial domain. For micro-eukaryotes, we considered all genomes assigned to the following manually selected genera: *Blastocystis, Candida, Saccharomyces, Cryptosporidium, Entamoeba, Aspergillus, Cryptococcus, Cyclospora, Cystoisospora, Giardia, Leishmania, Malassezia, Neosartorya, Pneumocystis, Toxoplasma, Trachipleistophora, Trichinella, Trichomonas,* and *Trypanosoma*.

Reference genomes ('fasta' format, suffix '.fna') and the associated genomic annotation ('GFF' format, suffix '.gff') of each proteome were downloaded from the NCBI GenBank FTP server (ftp.ncbi.nlm.nih.gov/genomes/all/GCA) by retrieving URLs from the assembly_summary_genbank.txt file (ftp.ncbi.nlm.nih.gov/genomes/genbank/assembly_summary_genbank.txt) using the GCA accession included in the UniProt Proteomes resource (01/24/2019). Starting from a total of 111,825 UniProt Proteomes entries, we discarded 12,598 proteomes missing the GenBank accession, ending up with 99,227 genomes (997 Archaea, 97,941 Bacteria, 339 Eukaryota).

The microbial proteins (and genes) associated to at least one UniProt Proteome and considered by ChocoPhlAn are retrieved from the UniProt Knowledgebase (UniProtKB) and the UniProt Archive (UniParc) databases. Proteins included in UniProtKB have been derived from the translation of the CDSs of all available reference genomes included in UniProt Proteomes. ChocoPhlAn 3 also retrieves and includes relevant data present in the UniProtKB entries (retrieved from ftp.uniprot.org/pub/databases/uniprot/ as XML files uniprot_sprot.xml.gz, uniprot_trembl.xml.gz, uniparc_all.xml.gz) such as functional, phylogenomic, and protein domain annotations (KEGG, KO, EggNOG, GO, EC, Pfam) (*El-Gebali et al., 2019*; *Huerta-Cepas et al., 2016*; *Kanehisa and Goto, 2000*; *The Gene Ontology Consortium, 2019*), accessions for cross-referencing entries with external databases (GenBank, ENA, and BioCyc) (*Clark et al., 2016*; *Karp et al., 2019*; *Leinonen et al., 2011*), name of the gene that encodes for the protein, and proteome accession.

We processed a total of 203.9M proteins included in both UniProtKB and UniParc, and 126.9M of them were associated with a UniProt Proteome entry. The Bacteria domain tallied the highest number of proteins (194.8M), whereas Archaea and Eukaryotes accounted for 5.0M and 4.0M proteins, respectively.

In order to reduce the redundancy of the database, we use the UniRef90 clustering of UniProtKB proteins provided by UniProt. In brief, UniProtKB are clustered at different thresholds of sequence identity (100, 90, 50) and made available through the UniProt Reference Clusters (UniRef) resource (*Suzek et al., 2015*). UniRef90 clusters are generated by clustering unique sequences (UniRef100, which combines identical UniProtKB proteins in a single cluster) via CD-HIT (*Li and Godzik, 2006*) until August 2019, and via MMseqs2 (*Steinegger and Söding, 2018*) afterward. Sequences in UniRef90 clusters have at least 90% sequence identity (*Suzek et al., 2015*). UniRef50 clusters are generated by clustering the UniRef90 cluster seed sequences, and each cluster contains proteins with at least 50% identity. Both UniRef90 and UniRef50 require each protein to overlap at least 80% with the cluster's longest sequence. UniRef entries considered in ChocoPhlAn 3 contain the sequence of a representative protein, the accession IDs of all the entries included in the cluster, the accessions to the UniProtKB and UniParc records, and the accessions of the other associated UniRef cluster are included in the UniProt entries.

A total of 292.1M UniRef clusters were processed (172.3M, 87.3M, and 32.5M for UniRef100, UniRef90, and UniRef50, respectively) and associated with each protein and each genome in ChocoPhlAn 3.

## Pan-proteome generation

We then generate pan-proteomes for each species represented at least by one UniProt Proteome. We define a species' pan-proteome as the non-redundant representation of the species' protein-coding potential. These are obtained for each species by considering the unique UniRef90 and UniRef50 protein families present in the genomes assigned at the species level and below.

For each pan-protein, we compute several scores. We define a 'coreness' score for a UniRef90 family as the number of genomes included in the species' pan-proteome having a protein belonging to the UniRef family, and the 'uniqueness' score as the number of pan-proteomes of other species possessing the same pan-protein. We then also considered a 'uniqueness_sp' score, a variant of the 'uniqueness' score obtained excluding those species that were previously tagged as low-quality species. Alongside the 'uniqueness' score, we compute the 'external_genomes' as the number of genomes (rather than species or species' pan-proteomes) of other species' pan-proteomes possessing the same pan-protein. These scores were computed for both UniRef50 and UniRef90 protein families.

In ChocoPhlAn 3 we consider a total of 22,096 species' pan-proteomes and a total of 87.3M UniRef90 core proteins (i.e. with coreness >0.7, avg. 3,952 s.d. 6311 per species).

## Generation of MetaPhlAn 3 markers

MetaPhlAn relies on a set of unique and species-specific nucleotide markers that were updated in MetaPhlAn 3 starting from the ChocoPhlAn 3 pan-proteomes. We initially filtered out species having taxonomies previously tagged as low quality using the species-level genome bin (SGB) system (*Pasolli et al., 2019*). 'Low-quality' species that were assigned to the same SGB were merged and only the representative SGB was taken into account.

This merging procedure occurred for a total of 1328 species (6%) that were merged as they were unlikely to be distinguishable in metagenomic samples and would potentially lead to false-positive taxonomic assignments (see *Supplementary file 7* for the merged species). For the cases in which multiple species included by the NCBI taxonomy into a 'species-group' showed a high number of markers with a high 'uniqueness' score (>30), we proceeded to identify unique markers for the whole species groups. This occurred for the following species groups: *Streptococcus anginosus* group, *Lactobacillus casei* group, *Bacillus subtilis* group, *Enterobacter cloacae* complex, *Pseudomonas syringae* group, *Pseudomonas stutzeri* group, *Pseudomonas putida* group, *Pseudomonas fluorescens* group, *Pseudomonas aeruginosa* group, *Streptococcus dysgalactiae* group, and *Bacillus cereus* group. In all these cases, the pangenomes were built by merging all the species-level pangenomes and treating them as a single species.

In the first step of the marker discovery procedure, we use the pan-proteome built using the UniRef90 clusters considering all proteins with a length between 150 and 1500 amino acids. Starting from the coreness and uniqueness scores, we applied an iterative approach in order to find up to 150 unique markers whenever possible and retaining only those species with a minimum of 10 unique markers. We classify candidate markers into unique and quasi-markers according to the 'uniqueness' value: markers having zero 'uniqueness' are reported as 'unique markers'. When no unique markers can be identified, the less-stringent thresholds used in the marker discovery procedure allows the identification of the so-called 'quasi-markers', markers having non-null values of 'uniqueness'.

The iterative approach started with the definition of four tiers of unique markers according to a combination of the values of 'coreness', 'uniqueness', and 'external_genomes'. Tier 'A' includes pan-proteins with a coreness score higher than 80%, not shared with more than two other pan-proteomes considering both UniRef90 and UniRef50 clustering score ('Uniqueness_NR90' and 'Uniqueness_NR50'), and not present in more than 10 single genomes when considering the UniRef90 and 5 single genomes when considering UniRef50 ('External_genomes_NR90' and 'External_genomes_NR50'), respectively. Tier 'B' includes markers with 'coreness' values between 70% and 80%, 'Uniqueness_NR90', and 'Uniqueness_NR50' values of 5, and values of 'External_genomes_NR90'

and 'External_genomes_NR50' lower than 15 and 10 genomes, respectively. Markers that did not meet the previous criteria were included in the 'C' tier, which includes markers with 'coreness' values between 50% and 70%, 'Uniqueness_NR90' less than 10, 'Uniqueness_NR50' less than 15, 'External_genomes_NR90' less than 25, and 'External_genomes_NR50' less than 20. Markers for the species having only one genome included in the pan-proteome, for which the definition of coreness is trivial, were classified as tier 'U', provided that they have zero 'Uniqueness'.

The definition of specific tiers allows the retrieval of the maximum number of unique markers. Marker discovery procedure was performed iteratively for each tier. Candidate markers that meet the tier-defined thresholds were ranked using a score function defined as follows:

$$Score = S_{coreness} * S_{uniqueness50} * S_{uniqueness90}$$

Where

$$S_{coreness} = \sqrt{coreness_\%}$$

$$S_{uniqueness90} = -log\left(1 - \frac{10^4 - min(10^4, uniqueness_{90})}{10^4 - 10^{-4}}\right) * \frac{1}{5}$$

$$S_{uniqueness50} = -log\left(1 - \frac{10^4 - min(10^4, uniqueness_{50})}{10^4 - 10^{-4}}\right) * \frac{1}{5}$$

The score function as defined will prioritize the selection of candidate markers highly conserved in the clade (high 'coreness' value) but shared with the smallest possible number of other species (low values of 'uniqueness'). Tier type is assigned to each candidate marker, and if more than 50 candidate markers were identified, we selected up to 150 markers from the ranked list. If not enough markers were identified (less than 50), the procedure was repeated using the subsequent tier's thresholds. If no markers were identified using tier C thresholds, the species was discarded.

Nucleotide sequences for each marker selected with this procedure are then considered as entries for the MetaPhlAn database. To refine the number of species estimated by the 'uniqueness' parameter, marker sequences were split into non-overlapping chunks of 150 bp and mapped versus an index built using all the reference genomes used for the marker identification process using bowtie2 (version 2.3.4.3, parameters '-a –very-sensitive –no-unal –no-hq –no-sq'). We accounted for a newly identified species based on the 'uniqueness' parameter if at least 150 consecutive nucleotides of the marker sequence were found in the identified target reference genome.

We performed an additional step of curation for markers for species with genomes obtained with Co-Abundance gene Groups (CAGs) (*MetaHIT Consortium et al., 2014*). To reduce the number of false positives, we removed the CAG species if more than 50% of its markers were shared with the species that gave the taxonomy to the CAG genome.

Each marker has associated an entry in the MetaPhlAn database which includes the species for which the sequence is a marker, the list of species sharing the marker, the sequence length, and the taxonomy of the species. Viral markers were taken from the v20_m200 MetaPhlAn 2 database.

Altogether, this identified a total of 1.1M markers for 13,475 species (*Supplementary file 8*).

## MetaPhlAn 3 taxonomic profiling

The raw reads in a metagenomic sample are mapped by MetaPhlAn 3 to a database of 1.1M markers using bowtie2 (*Langmead and Salzberg, 2012*). The default bowtie2 mapping parameters are those of the 'very-sensitive' preset but are customizable via the MetaPhlAn 3 settings. In MetaPhlAn 3, the input can be provided as a single FASTQ file (optionally compressed), multiple FASTQs in a single archive, or as a pre-performed mapping. Internally, MetaPhlAn 3 estimates the coverage of each marker and computes the clade's coverage as the robust average of the coverage across the markers of the same clade. The clade's coverages are then normalized across all detected clades to obtain the relative abundance of each taxon as previously described (*Segata et al., 2012*; *Truong et al., 2015*).

In version 3, we further optimized the parameter of the robust average which excludes the top and bottom quantiles of the marker abundances ('stat_q' parameter). This is now set by default to

0.2 (i.e. excludes the 20% of markers with the highest abundance as well as the 20% of markers with the lowest abundance). To further improve the quality of the read mapping, we adopted quality controls before and after mapping by discarding low-quality sequences and alignments (reads shorter than 70 bp and alignment with a MAPQ value less than 5).

We also introduced a new feature for estimating the 'unknown' portion of the taxonomic profile that would correspond with taxa not present in current databases; this is computed by subtracting from the total number of reads the average read depth of each taxon normalized by its taxon-specific average genome length. Additionally, the new output format for MetaPhlAn 3 by default includes the NCBI taxonomy ID of each profiled clade, allowing for better comparisons between tools and tracking of the species name in case of taxonomic reassignment.

Finally, alongside the default MetaPhlAn output format, profiles can be now reported using the CAMI output format defined by *Belmann et al., 2015*; *BioBoxes, 2020* that can be used for performing benchmarks with the OPAL framework (*Meyer et al., 2019*). To support post-profiling analyses, a convenience R script for computing weighted and unweighted UniFrac distances (*Lozupone and Knight, 2005*) from MetaPhlAn profiles is now available in the software repository (metaphlan/utils/calculate_unifrac.R), alongside the phylogeny (in Newick format) comprising all MetaPhlAn 3 taxa. The improvements and addition in MetaPhlAn 3 compared to the previous MetaPhlAn two version are summarized in *Supplementary file 2*.

## StrainPhlAn 3 strain profiling

StrainPhlAn performs genotyping at the strain level by reconstructing sample-specific consensus sequences of MetaPhlAn markers and using them for multiple-sequence alignment and phylogenetic modeling (*Truong et al., 2017*). StrainPhlAn 3 improves the original implementation in several aspects: (i) the integration of an improved and validated pipeline for consensus sequence generation (*Zolfo et al., 2019*), (ii) the integration of PhyloPhlAn 3 (*Asnicar et al., 2020*) which improves the quality of the phylogenetic modeling and the flexibility of the analysis, and (iii) a refined algorithm for filtering samples not supported by enough species' markers and markers not enough conserved across strains and samples.

StrainPhlAn 3 takes as input the alignment results from the MetaPhlAn 3 profiling (i.e. the mapping of the metagenomic samples against the MetaPhlAn species-specific markers) as well as the MetaPhlAn 3 markers' database. For each sample, StrainPhlAn 3 reconstructs high-quality consensus sequences of the species-specific markers by considering, at each position of the marker, the nucleotide with the highest frequency among the reads mapping against the marker and covering that position. By default, consensus markers reconstructed with less than eight reads or with a breadth of coverage (i.e. fraction of the marker covered by reads) lower than 80% are discarded ('–breadth_-threshold' parameter). Ambiguous bases are defined as positions in the alignment with quality lower than 30 or high polymorphisms (major allele dominance lower than 80%) and are considered for the threshold on the breadth of coverage as unmapped positions.

After marker reconstruction, the filtering algorithm discards samples with less than 20 markers, as well as markers present in less than 80% of the samples ('–sample_with_n_markers' and '–marker_-in_n_samples' parameters, respectively). Then, markers are trimmed by removing the leading and trailing 50 bases ('–trim_sequences' parameter), since these are usually supported by lower coverage due to the boundary effect during mapping, and a polymorphic rates report is generated for optional inspection by the user. Finally, filtered samples and markers are processed by PhyloPhlAn 3 for phylogenetic reconstruction. By default, reconstructed sequences are mapped against the markers database using BLASTn (*Altschul et al., 1990*), multiple sequence alignment is performed by MAFFT (*Katoh and Standley, 2013*) and phylogenetic trees are produced by RAxML (*Stamatakis, 2014*). Due to the reconstruction of a strain-level phylogeny, PhyloPhlAn was set to run with '–diversity low' parameter.

Phylogenetic trees produced by StrainPhlan 3 can also be used to identify identical strains across samples, which can be exploited, for example, in strain transmission analyses (*Ferretti et al., 2018*; *Shao et al., 2019*). This is now supported by the newly added 'strain_transmission.py' script. This script processes the phylogenetic tree produced by StrainPhAn together with metadata describing relations between the samples (e.g. longitudinal samples or samples with a relation of interest such as mother/infant pairings) to infer strain transmission events. First, using the phylogenetic tree, a pairwise distance matrix is generated and normalized by the total branch length of the tree. Using

the distance matrix and the associated metadata, a threshold defining identical strains is inferred selecting the first percentile of the distribution of the non-related-samples distances (i.e. setting an upper bound on the theoretical false-discovery rate at 1%). If longitudinal samples are provided, only one is considered per subject, and samples not included in the metadata are considered as non-related. Finally, related sample pairs with a distance smaller than the inferred threshold are reported as potential transmission events.

## HUMAnN 3 data and algorithm updates

Functional potential profiling of microbial communities is performed by HUMAnN using pange-nomes annotated with UniRef90 on all species detectable per sample with MetaPhlAn. ChocoPhlAn pangenomes used by HUMAnN for functional profiling are directly available as the species pan-pro-teomes annotated with the UniRef90 clusters. To obtain a nucleotide representation of each pan-proteome, we identified a representative of the cluster for each pan-protein by selecting a Uni-ProtKB or UniParc entry taxonomically assigned to the desired species. Each cluster representative was used for extracting the nucleotide sequence from the source reference genome and the several functional annotations from different systems (GO terms *Ashburner et al., 2000*), KEGG modules (*Kanehisa et al., 2014*), KO identifiers, Pfam accessions (*Finn et al., 2014*), EC numbers (*Bairoch, 2000*), and eggNOG accessions (*Powell et al., 2014*) associated with the UniProtKB entry. Alongside the functional annotations, we associated each UniRef90 cluster with its corresponding UniRef50 cluster in order to provide multiple levels of functional resolution.

HUMAnN 3 implements a number of new options for fine-tuning the steps in its tiered search (e.g. passing custom search parameters to bowtie2 *Langmead and Salzberg, 2012* and DIAMOND *Buchfink et al., 2015* in the pangenome and translated search steps, respectively). We performed a round of additional accuracy and performance tuning on these new parameters prior to the main evaluations of the paper (*Figure 1—figure supplement 4*). To minimize overfitting potential, we conducted initial tuning of HUMAnN 3 on the above-described human-like synthetic metagenome, which featured a structure and species composition that were distinct from those of the CAMI and nonhuman synthetic metagenomes used in downstream inter-method comparisons (*Figure 1*). We note that, because these tuning evaluations include parameter configurations that are equivalent to HUMAnN 2 defaults applied to the expanded bioBakery 3 databases, they quantify improvements to HUMAnN 3's accuracy and performance that are not attributable to its more complete database.

We first considered two new options when assigning reads to species pangenomes: (1) requiring pangene sequences to be covered above a threshold fraction of sites before any alignments to those sequences were accepted ('database sequence coverage filtering') and (2) allowing a read to align to multiple pangenes instead of the single target favored by bowtie2's default settings (as used in HUMAnN 2). Coverage filtering (new option 1) was already implemented in HUMAnN 2 for post-processing translated search results, where it was shown to increase UniRef90-level specificity considerably at a small cost to sensitivity (*Franzosa et al., 2018*). We observed similar results here in the context of pangenome search; as a result, HUMAnN 3 now imposes (separately tunable) database-sequence coverage filters during its pangenome and translated search steps (both default to 50%; *Figure 1—figure supplement 4*). Conversely, allowing a read to hit up to five pangenes (new option 2, as implemented via bowtie2's '-k 5' setting) had very little impact on accuracy and is not enabled by default in HUMAnN 3.

We additionally considered new options to tune the stringency and memory usage of DIAMOND 0.9 during translated search. The most impactful of these was reducing the identity threshold for per-read alignment to UniRef90 from 90% (the HUMAnN two default) to 80% (the new default for HUMAnN 3; *Figure 1—figure supplement 4*). While the former value was chosen to respect the average identity among UniRef90 family members, the 80% threshold is more forgiving of variation within read-length windows of a protein-level UniRef90 alignment. Coupled with HUMAnN's database sequence coverage filter, the 80% threshold correctly aligns considerably more reads during translated search without compromising specificity.

While HUMAnN 2 accepted DIAMOND's (default) top-20 database targets per query read, we newly evaluated the top 1 and top 5 targets, as well as any targets within 1, 2, or 10% of the best hit's score. We selected the 'within 1% score of the best hit' filter (DIAMOND's '–top 1' option) as a new default for HUMAnN 3 on the basis of a marked increase in UniRef90 specificity with minimal loss of sensitivity (*Figure 1—figure supplement 4*). Finally, we explored tuning DIAMOND's

memory via the '–block-size (-b)' and '–index-chunks (-c)' flags. We found the achievable increases in speed to be small relative to their corresponding memory requirements, and so HUMAnN 3 continues to favor DIAMOND's default, lower-memory configuration.

## PanPhlAn 3 with expanded databases and functional annotations

PanPhlAn performs strain-level metagenomic profiling by identifying the species-specific gene repertoire composition inside individual metagenomic samples (*Scholz et al., 2016*). It maps metagenomes against the pangenome of a species of interest using bowtie2 (*Langmead and Salzberg, 2012*). After coverage normalization (by summing the gene coverage of all genes in a gene family and dividing it by the average gene length of that family), PanPhlAn builds a coverage curve of genes' families across each sample and assesses which of these gene families are present or absent. This leads to the creation of a binary matrix of gene family presence/absence across all samples.

Compared to the previous versions, in PanPhlAn 3 we adopt a new ChocoPhlAn 3 pre-computed pangenome database of 2298 species built from species included in MetaPhlAn 3 for which at least two reference genomes are available. For species having more than 200 reference genomes available, the pangenome is made using a representative subset of 200 genomes maximizing the Mash distances between them (*Ondov et al., 2016*). PanPhlAn pangenomes from the database are composed of a FASTA file of all contigs, pre-computed bowtie2 indexes and a tab-separated values file containing the UniRef90 ID of the gene family as well as gene name, position in genomes, on contigs, and functional and structural annotations.

Moreover, new functionalities include a script for quick visualization of the presence/absence matrix with functionalities for clustering of gene family's profiles across samples. An empirical p-value can be computed for each cluster based on the ratio between the sum of the genes' lengths of one group and its total span along the contig. Thus, a significantly 'close' genes group can be identified and computation of empirical p-values assessing whether or not the genetic proximity of these families along the contigs could be considered significant. This eases the detection and identification of mobile elements in metagenomic samples.

## PhyloPhlAn 3

PhyloPhlAn 3 is an easy-to-use method to perform taxonomic contextualization and phylogenetic analysis of microbial genomes and of metagenome-assembled genomes (MAGs). PhyloPhlAn among its databases exploits both the set of core genes and of reference genomes identified by ChocoPhlAn 3 and extracted from the 111,825 UniProt Proteomes for each taxonomic species. The methods, performance, and examples of PhyloPhlAn are described elsewhere (*Asnicar et al., 2020*) and refers to the same version incorporated into bioBakery 3. In brief, the core genes included in the PhyloPhlAn 3 database are used to identify sequence homologs in the input genomes and MAGs that are then aligned, concatenated, and used for phylogeny reconstruction. A set of MAGs previously analyzed (*Pasolli et al., 2019*) can also be included to provide phylogenetic contextualization of newly assembled MAGs. PhyloPhlAn 3 thus provides the methodology to integrate assembly-based methods and phylogenetic analysis into the bioBakery 3 analysis framework.

## Synthetic metagenomes and gold standards for bioBakery 3 evaluations

We tuned and evaluated MetaPhlAn 3 and HUMAnN 3 using multiple different synthetic metagenomes of known species and gene content. The first set included synthetic metagenomes and gold-standard taxonomic profiles from the CAMI challenge representing five human body site-specific microbiomes and the murine gut microbiome (*Fritz et al., 2019*; *Sczyrba et al., 2017*). All such CAMI metagenomes were used for the evaluation of taxonomic profiling methods (including MetaPhlAn 3) while the first five lexically ordered metagenomes from each environment (human body sites and mouse gut) were used for the evaluation of functional profiling methods (including HUMAnN 3).

Second, because gold standard functional profiles were not provided for the CAMI metagenomes, we generated them ourselves by (1) functionally annotating the genomes sampled to build the CAMI metagenomes (and then 2) weighting their functional contributions according to mean coverage depth per 'sample'. Notably, this approach to gold-standard construction does not

account for gene-to-gene variation in read sampling along the length of community genomes. As a result, comparing the gold standards with functional profiles derived directly from the metagenome underestimates the profiles' accuracy (by ~0.1 units of Bray-Curtis distance at the UniRef90 level).

We applied procedures for community genome annotation developed during HUMAnN2 benchmarking to aid in gold-standard construction (*Franzosa et al., 2018*). Briefly, we first identified and translated open reading frames (ORFs) within the CAMI genomes using Prodigal (*Hyatt et al., 2010*), and then aligned the translated ORFs against the v3 UniRef90 and UniRef50 sequence databases using DIAMOND (*Buchfink et al., 2015*). Each ORF was assigned to the best-scoring UniRef90 family to which it aligned with at least 90% identity and 80% mutual coverage (if any); similarly, ORFs were assigned to the best-scoring UniRef50 family to which they aligned with at least 50% identity. Functional annotations were then transferred from UniRef90 and UniRef50 representatives to the corresponding ORFs, with UniRef90-derived, enzyme commission (EC) annotations forming the basis for the main functional profiling evaluation (*Figure 1* and *Figure 1—figure supplement 3*).

We constructed additional synthetic metagenomes by sampling sequencing reads from curated microbial genome sets using ART (*Huang et al., 2012*) with an Illumina HiSeq 2500 error model. One such group of metagenomes (abbreviated synphlan-nonhuman) was designed to mirror the sequencing depth and community structure of the CAMI metagenomes: that is inclusive of 30 million, 150-nt paired-end sequencing reads sampled from species genomes with a log-normal abundance distribution. However, the synphlan-nonhuman metagenomes are distinct from the CAMI metagenomes in that they exclude genomes of human-associated microbial species (defined as species detected in MetaPhlAn 3 profiles of metagenomes from the Expanded Human Microbiome Project, HMP1-II *Lloyd-Price et al., 2017*). In addition, 50% of species sampled for the synphlan-nonhuman metagenomes were associated with at least two sequenced isolate genomes and 50% of species pairs were congeneric sisters. We constructed an additional synthetic metagenome (synphlan-humanoid) based on the top-50 most abundant species detected from HMP1-II metagenomes to use for initial tuning of HUMAnN 3 (*Figure 1—figure supplement 4*). This metagenome contained 10 million, 100-nt paired-end reads sampled evenly from underlying species genomes. We constructed gold standard taxonomic profiles for these metagenomes based on the sampled genomes' taxonomic annotations and target sampling coverage; we constructed gold standard functional profiles based on UniProt-derived annotations of the species' protein-coding genes.

## Evaluation of MetaPhlAn 3 and HUMAnN 3 on synthetic data

To assess the performance of MetaPhlAn 3, we compared it with its previous version, MetaPhlAn 2 (*Truong et al., 2015*), alongside mOTUs2 (*Milanese et al., 2019*) and Bracken (*Lu et al., 2017*; *Wood et al., 2019*). We profiled a total of 118 synthetic metagenomes spanning different ecosystems: (i) 49 synthetic metagenomes (10 Airways, 10 Gastrointestinal Tract, 10 Oral, 10 Skin, 9 Urogenital tract) provided by the 2nd CAMI challenge (*Sczyrba et al., 2017*) resemble the composition of the Human Microbiome as described by the Human Microbiome Project *Turnbaugh et al., 2007*; (ii) 64 synthetic metagenomes generated by CAMISIM and modeled after the murine gut microbiome (*Fritz et al., 2019*); (iii) five synthetic metagenomes including non-human associated species (see above).

Each software was run using default parameters as described in their respective user manuals. Additionally, mOTUs2 was run with parameters '-C recall' and '-C precision' in order to increase precision and recall, respectively. When not directly available from the tool (MetaPhlAn 2 and Bracken), output profiles were converted into the CAMI output format as described by the BioBoxes RFC (*Belmann et al., 2015*; *BioBoxes, 2020*) in order to benchmark with the OPAL framework (*Meyer et al., 2019*) (version 1.0.5).

From the panel of measures computed by OPAL, we selected a subset (precision, recall, F1 score) for comparisons (*Supplementary file 9*). In addition to these measures, we computed the Pearson Correlation Coefficient between the predicted and expected relative abundance and the Bray-Curtis similarity index using arcsin square-root normalized relative abundances (*Supplementary file 3*).

MetaPhlAn 3 includes markers describing species groups, a case is not taken into account by OPAL. To perform the evaluation, we expanded the species group to represent all contained species and considered a true positive if the expected species matches one species taxonomically placed under the species group. In case of no matches, we consider as false positive only one species.

We also assessed the performance in terms of run-time and memory usage. We profiled five HMP samples (SRS014235, SRS011271, SRS064645, SRS023346, SRS048870) with all the aforementioned software (using only one thread) and tracked every second of the execution till the end of process the resident set size (RSS) memory usage using ps.

We evaluated HUMAnN 3, HUMAnN 2 (*Franzosa et al., 2018*), and Carnelian (*Nazeen et al., 2020*) on 30 CAMI metagenomes and the five synphlan-nonhuman metagenomes. Evaluations of HUMAnN 3 were carried out using version 3.0.0-alpha of the software, MetaPhlAn 3, bowtie2 version 2.3.5.1, and DIAMOND version 0.9.24. Evaluations on HUMAnN 2 were carried out using version 0.11.1 of the software, MetaPhlAn version 2.7.5, bowtie2 version 2.3.5.1, and DIAMOND version 0.8.36 (HUMAnN 2 is not compatible with DIAMOND version 0.9). HUMAnN 3 and 2 were run with their default settings and full-size databases alongside the '–threads 6' option. UniRef90 abundance profiles were converted to EC abundance profiles (to facilitate comparisons with Carnelian) using the 'uniref90_level4ec' option of the humann_regroup_table script.

We evaluated Carnelian version 1.0.0 following installation and usage instructions given at http://cb.csail.mit.edu/cb/carnelian/ and https://github.com/snz20/carnelian. Specifically, we first converted synthetic metagenome reads to FASTA format (this step was not counted toward the total runtime of the Carnelian method). Reads were then scanned for peptide fragments using 'carnelian.py translate' wrapping FragGeneScan (*Rho et al., 2010*) version 1.31 with the '-n 3' option. Peptides were then assigned to EC categories using 'carnelian.py predict' wrapping Vowpal Wabbit 8.1.1 and the EC-2010-DB model supplied at the above URLs. Finally, adjusted EC abundances were estimated using 'carnelian.py abundance' and the average EC family gene lengths supplied with the software and a fragment size of 150 (to match the reads of the CAMI and synphlan-nonhuman metagenomes).

All method calls were made with the humann_benchmark utility script to track total runtime and memory usage (maximum resident set size, MaxRSS). Runtimes were converted to equivalent CPU-hours. For multi-step computations, CPU-hours were summed while the overall maximum MaxRSS was retained. Predicted EC abundances were sum-normalized to 1.0 at the community and per-species levels prior to Bray-Curtis similarity computations.

## Colorectal cancer microbiome meta-analysis

We applied the new MetaPhlAn 3 and HUMAnN 3 on a set of human gut metagenomes profiling colorectal cancer patients and controls, updating our previous meta-analyses performed with MetaPhlAn 2 and HUMAnN 2 (*Thomas et al., 2019*; *Wirbel et al., 2019*). To the previous meta-analysis, we added two more datasets that became available afterward (*Gupta et al., 2019*; *Yachida et al., 2019*). In total, we analyzed 1262 metagenomes from 10 datasets (for a total of CRC metagenomes and 600 controls, *Supplementary file 10*). The dataset was stratified by country of origin with the exception of the two Italian cohorts published in *Thomas et al., 2019* which were kept separate due to differences in the DNA extraction protocols. Results were thus computed on nine distinct sub-cohorts.

MetaPhlAn 3 and HUMAnN 3 were used for the taxonomic and functional profiling of all sub-cohorts. Meta-analysis on the species-level, pathways, UniRef90 gene families, and enzyme commission (EC) categories relative abundances were performed on the sub-cohorts as previously described (*Thomas et al., 2019*). In brief, relative abundances were arcsine-square-root transformed, Cohen's D was computed by the escalc function (metafor R package *Viechtbauer, 2010*) to model random effects, and $I^2$ estimates and Cochran's Q-test were used for quantifying study-heterogeneity and assessing their statistical significance. Multidimensional scaling analysis was performed on the Weighted UniFrac distance (vegan 'cmdscale' and rbiom 'unifrac' function *Oksanen et al., 2008*) computed on the relative abundance data adjusted for study batch effect with MMUPHin (*Ma, 2019*) and normalized using arcsin-square root. Alpha-diversity analysis was performed on the data after being rarefied to the 10th percentile of the read depth in each cohort.

We used MetAML (*Pasolli et al., 2016*) to feed species-level and pathway-level relative abundances to a Random Forest model (*Breiman, 2001*). Age was also added to the feature-set, as this covariate has been shown to improve microbiome predictions in CRC (*Ghosh et al., 2020*). MetAML executed the Random-Forest implementation by Scikit-Learn v.0.22.2 with the following parameters: 10,000 estimator trees, square-root as the proportion of feature sampled in entrance to each estimator, no-maximum depth for the trees, one sample as the minimum amount for each leaf of each tree,

'gini' as impurity criterion. Considering each cohort, we tested the taxonomical and the functional potential profiles in the CRC prediction problem in a standard cohort-specific cross-validation as well as on the more reproducible leave-one-dataset-out (LODO) setting (*Thomas et al., 2019*; *Wirbel et al., 2019*).

UniRef90 *cutC* gene family IDs were selected from the UniRef90 database included in HUMAnN 3. Species richness was calculated by tallying species with non-zero relative abundance. Differential species richness and *cutC* abundance tests were performed using the Wilcoxon rank-sum test, wilcox.test, as implemented in the 'stats' R package.

## HMP2 IBD metagenome and metatranscriptome profiling

We applied MetaPhlAn 3 and HUMAnN 3 to 1635 metagenomes and 817 metatranscriptomes from the HMP2 Inflammatory Bowel Disease (IBD) Multi-omics Database (IBDMDB) (*IBDMDB Investigators et al., 2019*). We took advantage of previously quality-controlled sequencing data from this cohort as downloaded from http://ibdmdb.org (June 2020). Following the standard bioBakery workflow (*McIver et al., 2018*) for combined meta-omic sequencing data, we processed the HMP2 metagenomes using HUMAnN 3.0.0.alpha.1 (including taxonomic prescreening performed by MetaPhlAn 3). We then processed the paired HMP2 metatranscriptomes using their corresponding metagenomic taxonomic profiles as guides for pangenome selection. To quantify improved performance in bioBakery 3, we compared the HUMAnN logs produced during the runs described above with logs downloaded from http://ibdmdb.org describing analyses of the same samples using MetaPhlAn 2.6.0 and HUMAnN 2.11.0.

To identify expression-level microbial metabolic biomarkers of IBD activity from the HMP2 dataset, we sum-normalized UniRef90 gene family abundance profiles to 'copies per million' (CPM) units and then summed UniRef90 CPMs according to enzyme commission (EC) annotations using HUMAnN utility scripts. We then compared community-level EC expression with other sample properties using a mixed effects model implemented in R's lmerTest package (*Kuznetsova et al., 2017*) (using subject as a random effect to account for repeated longitudinal sampling):

$$log(RNA) \sim log(DNA) + diagnosis + diagnosis{:}active + age + antibiotics + (1|subject)$$

For a given EC, we evaluated the above model over paired meta-omes in which the EC's metatranscriptomic abundance (RNA) and metagenomic abundance (DNA) were both non-zero; ECs were excluded if they failed to satisfy this condition in at least 10% of paired meta-omes. This approach avoids interpreting RNA non-detection as strong evidence of 'down-regulation' (relative to DNA abundance, identifying zero RNA reads for a feature is more common due to the wide dynamic range of gene expression values and the large fraction of sequencing depth absorbed by non-coding RNAs).

The inclusion of DNA abundance as a covariate in the above model accounts for the strong dependence between a function's gene (metagenomic) copy number and its metatranscriptomic abundance. Thus, associations between EC RNA and other covariates can be interpreted as associations with 'residual expression' (potentially reflecting up- or down-regulation of community genes independent of changes in metagenome structure). Subject age at study enrollment and per-sample antibiotics exposure were included as additional clinical covariates. The statistical significance of model covariates was assessed after performing Benjamini-Hochberg FDR correction on model p-values batched by covariate and level.

We focused on associations between residual EC expression and subject diagnosis and disease activity with diagnosis. Here, subject diagnosis was divided broadly into Crohn's disease (CD; n = 49), ulcerative colitis (UC; n = 30), and non-IBD controls (n = 27). Due to the longitudinal nature of the HMP2 dataset, subjects diagnosed with CD and UC experienced variation in disease severity over the course of the study. The effects of disease activity on the microbiome were previously quantified as a 'dysbiosis score' (*IBDMDB Investigators et al., 2019*) measuring ecological deviation from the control microbiome population. Samples from CD and UC patients that deviated most strongly by this measure were classified as 'active.' Of 788 paired meta-omes considered here, 363 were from CD patients (76 with 'active' CD), 227 were from UC patients (23 with 'active' UC'), and 198 were from non-IBD controls. Consistent with earlier analyses of the HMP2 dataset (*IBDMDB Investigators et al., 2019*), we did not detect significant differences in EC expression as a

function of diagnosis alone (i.e. independent of disease activity), as non-active IBD meta-omes tend to be similar to those from control patients.

## Strain-level analysis of *Ruminococcus bromii*

For *Ruminococcus bromii* population genetic analysis, from the 9316 metagenomes spanning 46 datasets considered by *Pasolli et al., 2019*, we selected 4077 samples in which *R. bromii* was found present with a relative abundance above 0.05%. Strain-level profiling with StrainPhlAn 3 was performed using default parameters. 702 samples were discarded due to the low number and/or poor quality of the reconstructed markers (samples having less than 20 markers and markers present in less than the 80% of the samples are excluded). A total of 124 *R. bromii* MetaPhlAn 3 markers were used to generate a multiple sequence alignment. A phylogenetic distance matrix was produced by the dismat function from the EMBOSS package (*Rice et al., 2000*) (Kimura 2-parameter distance correction) using the multiple sequence alignment file produced by StrainPhlAn. Prediction strength analysis performed on the phylogenetic distance matrix using the prediction.strength function included in the 'fpc' R package (*Hennig, 2010*) version 2.2 revealed the presence of 4 optimal clusters (strength threshold 0.8). PAM clustering was subsequently applied on the phylogenetic distance matrix using the 'cluster' R package (*Kaufman and Rousseeuw, 2009*) version 2.1. The phylogenetic tree generated by PhyloPhlAn was plotted with GraPhlAn (*Asnicar et al., 2015*). For visualization purposes, samples were grouped by continent with the exception of the Chinese cohorts. Tree cluster colors were assigned by considering the most common cluster assigned to leaves, and clusters 3 and 4 were joined into the 'Others' group for the sake of discussion. In order to detect possible events of vertical transmission of *R. bromii*, we executed the 'strain_transmission.py' script using as input the phylogenetic tree produced by StrainPhlAn.

Pangenome-based strain-level analysis was performed on the same selected set of samples using PanPhlAn 3 with the *R. bromii* pangenome composed of 8 reference genomes available on NCBI (GCA_002834165, GCA_002834225, GCA_002834235, GCA_003466165, GCA_003466205, GCA_003466225, GCA_900101355, and GCA_900291485). After mapping the metagenomic samples to the pangenome, a binary matrix of presence/absence was built using the PanPhlAn profiling script with default options for strain detection and filtering (`–min_coverage 2 –left_max 1.25 –right_min 0.75`). The resulting matrix describes the presence/absence of 6,151 UniRef90 families across 2679 metagenomics samples and eight reference genomes.

In order to simplify the visualization of these results, we first discarded the genes families present in less than two samples or absent in five or less samples. Then, the Jaccard distance based on presence/absence fingerprint was computed for both genes families and samples. Hierarchical clustering was built using the Ward criterion ('ward.D2' in R 'hclust' function). A second more stringent filtering removed all genes families present in more than 95% or less than 5% of the remaining samples.

For assessing the correlation between the strain-level genomics and pangenomics results, we compared the phylogenetic distance distributions retrieved from the StrainPhlAn and PanPhlAn analyses. We used RAxML version 8.2.4 (*Stamatakis, 2014*) to generate phylogenetic distances between samples from PanPhlAN results. PanPhlAn information was coded as the presence-absence fingerprint of each sample and distances were computed using the substitution model based on these two states (argument -m MULTICAT of RAxML). One outlier sample was discarded due to mislabeled genomes. The StrainPhlAn phylogenetic distances were produced during the execution of the 'strain_transmission.py' script. Correlation between PanPhlAn and StrainPhlAn pairwise distances was calculated using the Pearson correlation Coefficient.

## Acknowledgements

The work was supported by the European Research Council (ERC-STG project MetaPG-716575) to NS; by MIUR 'Futuro in Ricerca' (grant No. RBFR13EWWI_001) to NS; by the European H2020 program (ONCOBIOME-825410 project and MASTER-818368 project) to NS; by the National Cancer Institute of the National Institutes of Health (1U01CA230551) to NS; by the Premio Internazionale Lombardia e Ricerca 2019 to NS; by the Harvard Chan Microbiome Analysis Core (CH); by the National Institute of Diabetes and Digestive and Kidney Diseases of the National Institutes of Health (R24DK110499 and U54DE023798) to CH; by Cancer Research UK Grand Challenge award C10674/A27140 to Wendy Garrett (CH); by the Juvenile Diabetes Research Foundation (3-SRA-2016–141-Q-

R) to CH; and by the National Human Genome Research Institute of the National Institutes of Health (R01HG005220) to Raphael Irizarry (CH).

## Additional information

### Funding

| Funder | Grant reference number | Author |
| --- | --- | --- |
| H2020 European Research Council | 716575 | Nicola Segata |
| Ministero dell'Istruzione, dell'Università e della Ricerca | RBFR13EWWI_001 | Nicola Segata |
| H2020 Health | 825410 | Nicola Segata |
| H2020 Food | 818368 | Nicola Segata |
| National Institutes of Health | 1U01CA230551 | Curtis Huttenhower |
| National Institute of Diabetes and Digestive and Kidney Diseases | R24DK110499 and U54DE023798 | Curtis Huttenhower |
| Cancer Research UK | C10674/A27140 | Curtis Huttenhower |
| Juvenile Diabetes Research Foundation United States of America | 3-SRA-2016-141-Q-R | Curtis Huttenhower |
| National Human Genome Research Institute | R01HG005220 | Curtis Huttenhower |

The funders had no role in study design, data collection and interpretation, or the decision to submit the work for publication.

### Author contributions

Francesco Beghini, Resources, Software, Validation, Methodology, Writing - original draft, Writing - review and editing; Lauren J McIver, Resources, Software, Validation, Writing - review and editing; Aitor Blanco-Míguez, Resources, Software, Visualization, Writing - review and editing; Leonard Dubois, Resources, Software, Visualization; Francesco Asnicar, Resources, Software, Validation, Methodology; Sagun Maharjan, Ana Mailyan, Moreno Zolfo, Resources, Software, Methodology; Paolo Manghi, Resources, Data curation, Investigation, Methodology; Matthias Scholz, Software; Andrew Maltez Thomas, Resources, Software, Validation, Investigation, Methodology; Mireia Valles-Colomer, Resources, Validation, Investigation, Methodology; George Weingart, Yancong Zhang, Resources, Software; Curtis Huttenhower, Conceptualization, Resources, Supervision, Funding acquisition, Methodology, Project administration, Writing - review and editing; Eric A Franzosa, Conceptualization, Resources, Software, Supervision, Methodology, Writing - original draft; Nicola Segata, Conceptualization, Resources, Supervision, Funding acquisition, Investigation, Methodology, Writing - original draft, Project administration, Writing - review and editing

### Author ORCIDs

Francesco Beghini ![ORCID] https://orcid.org/0000-0002-8105-9607
Leonard Dubois ![ORCID] http://orcid.org/0000-0003-0322-482X
Matthias Scholz ![ORCID] http://orcid.org/0000-0003-1414-5924
Yancong Zhang ![ORCID] http://orcid.org/0000-0002-2768-2975
Moreno Zolfo ![ORCID] http://orcid.org/0000-0001-6661-4046
Curtis Huttenhower ![ORCID] https://orcid.org/0000-0002-1110-0096
Eric A Franzosa ![ORCID] https://orcid.org/0000-0002-8798-7068
Nicola Segata ![ORCID] https://orcid.org/0000-0002-1583-5794

### Decision letter and Author response
Decision letter https://doi.org/10.7554/eLife.65088.sa1

Author response https://doi.org/10.7554/eLife.65088.sa2

## Additional files

### Supplementary files

• Supplementary file 1. Average values of F1 scores of MetaPhlAn 3, MetaPhlAn 2, mOTUs2, and Kraken species-level profiles computed on the 123 synthetic metagenomes.

• Supplementary file 2. bioBakery 3 software improvements.

• Supplementary file 3. Mean and ranked values of Bray-Curtis similarity and arcsine-square-root normalized Bray-Curtis similarity obtained by MetaPhlAn 3, MetaPhlAn 2, mOTUs2, and Kraken on the synthetic metagenomes considered in the evaluation.

• Supplementary file 4. Comparison of runtime and memory consumption of MetaPhlAn 3, MetaPhlAn 2, mOTUs2, and Kraken + Bracken on the 5 HMP metagenomes.

• Supplementary file 5. MetaPhlAn 3 taxonomic profiles and HUMAnN 3 functional profiles of the 1262 CRC samples.

• Supplementary file 6. MetaPhlAn 3 species-level, HUMAnN 3 pathway abundances and gene families abundances CRC meta-analysis results.

• Supplementary file 7. MetaPhlAn 3 species merged according to the species-level genome bin (SGB) system.

• Supplementary file 8. Number of distinct MetaPhlAn 3 markers per species.

• Supplementary file 9. Per-sample OPAL binary measures (true positive, false positive, false negative, precision, recall, F1 score) computed on MetaPhlAn 3, MetaPhlAn 2, mOTUs2, and Kraken species-level profiles computed on the 123 synthetic metagenomes.

• Supplementary file 10. Metadata of all the 1262 samples from the 10 CRC datasets.

• Transparent reporting form

### Data availability

Human and murine synthetic metagenomes and gold standards provided by the CAMI Challenge are available at https://data.cami-challenge.org/participate. Non-human synthetic metagenomes and gold standards are available at http://segatalab.cibio.unitn.it/tools/biobakery/. CRC metagenomic datasets analyzed in the meta-analysis are available in BioProject under accession numbers PRJEB7774, PRJNA531273, PRJNA447983, PRJDB4176, PRJEB12449, PRJEB27928, PRJDB4176, PRJEB10878, and PRJEB6070. Sequences and data for the Integrative Human Microbiome Project are available at the IBDMDB website (https://ibdmdb.org/) and deposited in SRA under accession number PRJNA398089. Taxonomic profiles, functional profiles, and sample metadata of the CRC datasets are available as Table S5 and Table S10. Taxonomic profiles and functional profiles of the HMP IBDMDB dataset are newly available at https://ibdmdb.org/. Profiles are also available through the curatedMetagenomicData R package (Pasolli et al., 2017). The full list of metagenomic datasets and samples used for the strain-level analysis of Ruminococcus bromii is reported in Table S1 from (Pasolli et al., 2019). Ruminococcus bromii reference genomes are deposited in GenBank under accession GCA_002834165, GCA_002834225, GCA_002834235, GCA_003466165, GCA_003466205, GCA_003466225, GCA_900101355 and GCA_900291485.

The following previously published datasets were used:

| Author(s) | Year | Dataset title | Dataset URL | Database and Identifier |
|---|---|---|---|---|
| Andrew Maltez Thomas, Paolo Manghi, Francesco Asnicar, Edoardo Pasolli, Federica Armanini, Moreno Zolfo, Francesco Beghini, Serena | 2018 | Whole-metagenome shotgun sequencing of two Italian fecal CRC cohorts | https://www.ncbi.nlm.nih.gov/bioproject/PRJNA447983 | BioProject, PRJNA447983 |

| | | | | |
|---|---|---|---|---|
| Manara, Nicolai Karcher, Chiara Pozzi, Sara Gandini, Davide Serrano, Sonia Tarallo, Antonio Francavilla, Gaetano Gallo, Mario Trompetto, Giulio Ferrero, Sayaka Mizutani, Hirotsugu Shiroma, Satoshi Shiba, Tatsuhiro Shibata, Shinichi Yachida, Takuji Yamada, Jakob Wirbel, Petra Schrotz-King, Cornelia M. Ulrich, Hermann Brenner, Manimozhiyan Arumugam, Peer Bork, Georg Zeller, Francesca Cordero, Emmanuel Dias-Neto, João Carlos Setubal, Adrian Tett, Barbara Pardini, Maria Rescigno, Levi Waldron, Alessio Naccarati, Nicola Segata | | | | |
| Fritz A, Hofmann P, Majda S, Dahms E, Dröge J, Fiedler J, Lesker TR | 2019 | CAMISIM: simulating metagenomes and microbial communities | https://data.cami-challenge.org/participate | CAMI Challenge, CAMISIM |
| Qiang Feng, Suisha Liang, Huijue Jia, Andreas Stadlmayr, Longqing Tang, Zhou Lan, Dongya Zhang, Huihua Xia, Xiaoying Xu, Zhuye Jie, Lili Su, Xiaoping Li, Xin Li, Junhua Li, Liang Xiao, Ursula Huber-Schönauer, David Niederseer, Xun Xu, Jumana Yousuf Al-Aama, Huanming Yang, Jian Wang, Karsten Kristiansen, Manimozhiyan Arumugam, Herbert Tilg, Christian Datz, Jun Wang | 2015 | Gut microbiome development along the colorectal adenoma–carcinoma sequence | https://www.ncbi.nlm.nih.gov/bioproject/PRJEB7774 | Bioproject, PRJEB7774 |
| Ankit Gupta, Darshan B. Dhakan, Abhijit Maji, Rituja Saxena, Vishnu Prasoodanan P.K., Shruti Mahajan, Joby Pulikkan, Jacob Kurian, Andres M. Gomez, Joy Scaria, Katherine R. Amato, Ashok K. Sharma, Vineet K. Sharma | 2019 | Association of Flavonifractor plautii, a Flavonoid-Degrading Bacterium, with the Gut Microbiome of Colorectal Cancer Patients in India | https://www.ncbi.nlm.nih.gov/bioproject/PRJNA531273 | BioProject , PRJNA531273 |

| | | | | |
|---|---|---|---|---|
| Emily Vogtmann, Xing Hua, Georg Zeller, Shinichi Sunagawa, Anita Y. Voigt, Rajna Hercog, James J. Goedert, Jianxin Shi, Peer Bork, Rashmi Sinha | 2016 | Colorectal cancer and the human gut microbiome | https://www.ncbi.nlm.nih.gov/bioproject/PRJEB12449 | BioProject , PRJEB12449 |
| Jakob Wirbel, Paul Theodor Pyl, Ece Kartal, Konrad Zych, Alireza Kashani, Alessio Milanese, Jonas S. Fleck, Anita Y. Voigt, Albert Palleja, Ruby Ponnudurai, Shinichi Sunagawa, Luis Pedro Coelho, Petra Schrotz-King, Emily Vogtmann, Nina Habermann, Emma Niméus, Andrew M. Thomas, Paolo Manghi, Sara Gandini, Davide Serrano, Sayaka Mizutani, Hirotsugu Shiroma, Satoshi Shiba, Tatsuhiro Shibata, Shinichi Yachida, Takuji Yamada, Levi Waldron, Alessio Naccarati, Nicola Segata, Rashmi Sinha, Cornelia M. Ulrich, Hermann Brenner, Manimozhiyan Arumugam, Peer Bork, Georg Zeller | 2019 | Meta-analysis of fecal metagenomes reveals global microbial signatures that are specific for colorectal cancer | https://www.ncbi.nlm.nih.gov/bioproject/PRJEB27928 | BioProject , PRJEB27928 |
| Shinichi Yachida, Sayaka Mizutani, Hirotsugu Shiroma, Satoshi Shiba, Takeshi Nakajima, Taku Sakamoto, Hikaru Watanabe, Keigo Masuda, Yuichiro Nishimoto, Masaru Kubo, Fumie Hosoda, Hirofumi Rokutan, Minori Matsumoto, Hiroyuki Takamaru, Masayoshi Yamada, Takahisa Matsuda, Motoki Iwasaki, Taiki Yamaji, Tatsuo Yachida, Tomoyoshi Soga, Ken Kurokawa, Atsushi Toyoda, Yoshitoshi Ogura, Tetsuya Hayashi, Masanori Hatakeyama, Hitoshi Nakagama, Yutaka Saito, Shinji Fukuda, Tatsuhiro | 2019 | Metagenomic and metabolomic analyses reveal distinct stage-specific phenotypes of the gut microbiota in colorectal cancer | https://www.ncbi.nlm.nih.gov/bioproject/PRJDB4176 | BioProject , PRJDB4176 |

| | | | | |
|---|---|---|---|---|
| Shibata, Takuji Yamada | | | | |
| Jun Yu, Qiang Feng, Sunny Hei Wong, Dongya Zhang, Qiao yi Liang, Youwen Qin, Longqing Tang, Hui Zhao, Jan Stenvang, Yanli Li, Xiaokai Wang, Xiaoqiang Xu, Ning Chen, William Ka Kei Wu Jumana Al-Aama, Hans Jørgen Nielsen, Pia Kiilerich, Benjamin Anderschou Holbech Jensen, Tung On Yau Zhou Lan, Huijue Jia, Junhua Li, Liang Xiao, Thomas Yuen Tung Lam Siew Chien Ng Alfred Sze-Lok Cheng Vincent Wai-Sun Wong Francis Ka Leung Chan Xun Xu, Huanming Yang, Lise Madsen, Christian Datz, Herbert Tilg, Jian Wang, Nils Brünner, Karsten Kristiansen, Manimozhiyan Arumugam, Joseph Jao-Yiu Sung Jun Wang | 2015 | Metagenomic analysis of fecal microbiome as a tool towards targeted non-invasive biomarkers for colorectal cancer | https://www.ncbi.nlm.nih.gov/bioproject/PRJEB10878 | BioProject , PRJEB10878 |
| Georg Zeller, Julien Tap, Anita Y Voigt, Shinichi Sunagawa, Jens Roat Kultima, Paul I Costea, Aurélien Amiot, Jürgen Böhm, Francesco Brunetti, Nina Habermann, Rajna Hercog, Moritz Koch, Alain Luciani, Daniel R Mende, Martin A Schneider, Petra Schrotz-King, Christophe Tournigand, Jeanne Tran Van Nhieu, Takuji Yamada, Jürgen Zimmermann, Vladimir Benes, Matthias Kloor, Cornelia M Ulrich, Magnus von Knebel Doeberitz, Iradj Sobhani, Peer Bork | 2014 | Colorectal cancer detection from fecal microbiota | https://www.ncbi.nlm.nih.gov/bioproject/PRJEB6070 | BioProject, PRJEB6070 |
| Indrani Mukhopadhya, Sarah Moraïs, Jenny Laverde-Gomez, | 2017 | ASM283416v1 | https://www.ncbi.nlm.nih.gov/assembly/GCF_002834165.1/ | GenBank , GCA_002834165 |

| | | | | | |
|---|---|---|---|---|---|
| Paul O. Sheridan, Alan W. Walker, William Kelly, Athol V. Klieve, Diane Ouwerkerk, Sylvia H. Duncan, Petra Louis, Nicole Koropatkin, Darrell Cockburn, Ryan Kibler, Philip J. Cooper, Carlos Sandoval, Emmanuelle Crost, Nathalie Juge, Edward A. Bayer, Harry J. Flint | | | | | |
| Indrani Mukhopadhya, Sarah Moraïs, Jenny Laverde-Gomez, Paul O. Sheridan, Alan W. Walker, William Kelly, Athol V. Klieve, Diane Ouwerkerk, Sylvia H. Duncan, Petra Louis, Nicole Koropatkin, Darrell Cockburn, Ryan Kibler, Philip J. Cooper, Carlos Sandoval, Emmanuelle Crost, Nathalie Juge, Edward A. Bayer, Harry J. Flint | 2017 | ASM283422v1 | | https://www.ncbi.nlm.nih.gov/assembly/GCA_002834225/ | GenBank , GCA_002834225 |
| Indrani Mukhopadhya, Sarah Moraïs, Jenny Laverde-Gomez, Paul O. Sheridan, Alan W. Walker, William Kelly, Athol V. Klieve, Diane Ouwerkerk, Sylvia H. Duncan, Petra Louis, Nicole Koropatkin, Darrell Cockburn, Ryan Kibler, Philip J. Cooper, Carlos Sandoval, Emmanuelle Crost, Nathalie Juge, Edward A. Bayer, Harry J. Flint | 2017 | ASM283423v1 | | https://www.ncbi.nlm.nih.gov/assembly/GCA_002834235/ | GenBank , GCA_002834235 |
| Yuanqiang Zou, Wenbin Xue, Guangwen Luo, Ziqing Deng, Panpan Qin, Ruijin Guo, Haipeng Sun, Yan Xia, Suisha Liang, Ying Dai, Daiwei Wan, Rongrong Jiang, Lili Su, Qiang Feng, Zhuye Jie, Tongkun Guo, Zhongkui Xia, Chuan Liu, | 2018 | ASM346616v1 | | https://www.ncbi.nlm.nih.gov/assembly/GCA_003466165/ | GenBank , GCA_003466165 |

| | | | | |
|---|---|---|---|---|
| Jinghong Yu, Yuxiang Lin, Shanmei Tang, Guicheng Huo, Xun Xu, Yong Hou, Xin Liu, Jian Wang, Huanming Yang, Karsten Kristiansen, Junhua Li, Huijue Jia, Liang Xiao | | | | |
| Yuanqiang Zou, Wenbin Xue, Guangwen Luo, Ziqing Deng, Panpan Qin, Ruijin Guo, Haipeng Sun, Yan Xia, Suisha Liang, Ying Dai, Daiwei Wan, Rongrong Jiang, Lili Su, Qiang Feng, Zhuye Jie, Tongkun Guo, Zhongkui Xia, Chuan Liu, Jinghong Yu, Yuxiang Lin, Shanmei Tang, Guicheng Huo, Xun Xu, Yong Hou, Xin Liu, Jian Wang, Huanming Yang, Karsten Kristiansen, Junhua Li, Huijue Jia, Liang Xiao | 2018 | ASM346620v1 | https://www.ncbi.nlm.nih.gov/assembly/GCA_003466205/ | GenBank , GCA_003466205 |
| Yuanqiang Zou, Wenbin Xue, Guangwen Luo, Ziqing Deng, Panpan Qin, Ruijin Guo, Haipeng Sun, Yan Xia, Suisha Liang, Ying Dai, Daiwei Wan, Rongrong Jiang, Lili Su, Qiang Feng, Zhuye Jie, Tongkun Guo, Zhongkui Xia, Chuan Liu, Jinghong Yu, Yuxiang Lin, Shanmei Tang, Guicheng Huo, Xun Xu, Yong Hou, Xin Liu, Jian Wang, Huanming Yang, Karsten Kristiansen, Junhua Li, Huijue Jia, Liang Xiao | 2018 | ASM346622v1 | https://www.ncbi.nlm.nih.gov/assembly/GCA_003466225/ | GenBank , GCA_003466225 |
| William Kelly | 2017 | IMG-taxon 2593339225 annotated assembly | https://www.ncbi.nlm.nih.gov/assembly/GCA_900101355/ | GenBank , GCA_900101355 |
| Indrani Mukhopadhya, Sarah Moraïs, Jenny Laverde-Gomez, Paul O. Sheridan, Alan W. Walker, William Kelly, Athol V. Klieve, Diane Ouwerkerk, Sylvia H. Duncan, Petra | 2018 | R_bromii_L2_63_validated_080218.embl.gz | https://www.ncbi.nlm.nih.gov/assembly/GCA_900291485/ | GenBank , GCA_900291485 |

Louis, Nicole
Koropatkin, Darrell
Cockburn, Ryan
Kibler, Philip J.
Cooper, Carlos
Sandoval,
Emmanuelle Crost,
Nathalie Juge,
Edward A. Bayer,
Harry J. Flint

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
