## [Decision Letter]

**Acceptance summary:**

Beghini and co-authors here present bioBakery 3, a platform that integrates a number of software tools for carrying out taxonomic and functional profiling of microbial communities from a wide range of environments. Together, these tools will accelerate microbiome research and be of broad utility to researchers across many fields.

**Decision letter after peer review:**

Thank you for submitting your article "Integrating taxonomic, functional, and strain-level profiling of diverse microbial communities with bioBakery 3" for consideration by *eLife*. Your article has been reviewed by 3 peer reviewers, one of whom is a member of our Board of Reviewing Editors, and the evaluation has been overseen by a Senior Editor. The following individual involved in review of your submission has agreed to reveal their identity: C Titus Brown (Reviewer #3).

The reviewers have discussed the reviews with one another and the Reviewing Editor has drafted this decision to help you prepare a revised submission.

Summary:

Beghini and co-authors here present bioBakery 3, a platform that integrates a number of software tools for carrying out taxonomic and functional profiling of microbiota from a wide range of environments. These tools are already well-established in the community, some even with their third versions publicly available. The authors describe updates to ChocoPhlan, MetaPhlan, HUMAnN, PhyloPhlan, StrainPhlan and PanPhlan, which all carry out specific tasks for analysing shotgun metagenome and metatranscriptome data. Perhaps the most significant improvement is related to a significantly updated database of reference sequences (ChocoPhlan), although algorithmic updates also allow for faster read mapping. It would be useful with more information on to what degree the updated database is responsible for the improved performance, and whether users are able to update this themselves.

The (many) benchmarking results and data analyses presented are of high caliber. For example, MetaPhlAn3 and HUMAnN are benchmarked using the state-of-the-art CAMI and perform well. We also applaud the inclusion of memory and CPU time numbers in the results.

The utility and versatility of the bioBakery tools is further demonstrated by deeper investigations into CRC and IBD metagenomes/transcriptomes, identifying a comparatively large number of potentially disease-associated taxa and genes, respectively. These include not previously reported oral species and TMA-producing genes that may be linked to CRC. Further studies based on complementary datasets and methodologies will be needed to confirm these.

Overall, the comparative tests and methodology descriptions are well described and supported by submitted data. The bioBakery 3 updates should improve meta-omic analysis and become a very useful resource for microbiome studies of a wide range of environments.

Essential revisions:

This seems like an excellent software package that could be further improved through the following additions:

1. The validation looks impressive but is unfortunately qualitative throughout. I would like to see the appropriate statistics included for all of the relevant contrasts (e.g., Figures 1b-d, Pg. 12 final paragraph, etc.) used to validate these methods throughout the manuscript.

2. It remains unclear why the performance of these different tools has been improved. Multiple ideas are mentioned in the text, including a more extensive database, improvements to parts of the algorithms, etc. The paper and field would benefit from a more targeted analysis to test what aspect of the update mattered most or alternatively if the improved performance is the aggregate result of a lot of little changes. At a minimum, it seems important to test how much these improvements are due to the expanded database as opposed to the tool itself. Could you get the same result with the version 2 tools and the up-to-date set of [meta]genomes?

MetaPhlan 3 has clearly improved precision compared to the 2nd versions. It would be useful to know to what degree this is solely due to the updated ChocoPhlan database. A couple of algorithm improvements are mentioned. This would be possible with the introduction of "Metaphlan 3 with db 2 (or 2.7) ", at least in a subset of the comparisons in displayed in Figure 1? This should answer what role the algorithmic improvements played.

3. The figures and legends could use some polishing to enable clarity. For example, Figure 1d shows multiple bars/box plots, but it's unclear what they represent due to the lack of x axis labels. Figure 2a includes lots of different colors indicating study that are impossible to distinguish due to the massive number of red samples, the legend to Figure 2c is missing, and the countries in Figure 4a are impossible to see any pattern.

4. The use cases of these updated tools are underwhelming and not clearly compared to the prior literature. The CRC analysis shown in Figure 2 highlight multiple bacterial species, one α diversity metric, and one gene (cutC) distinct from the healthy controls in this meta-analysis. It remains unclear which of these findings are new and if they are new, whether or not the new findings are better or worse than what was previously published. Similar issues affect the re-analysis of the IBD data (Figure 3) and the pan-genome data (Figure 4). While these figures provide a nice example of the analyses that can be done, it's unclear if anything new has been learned and if bioBakery 3 was necessary to run these analyses. These analyses also take up the majority of the main figures (3 out of 4), distracting from the main goal of the paper, which is to explain the improvements that have been made and to compare these updated tools to their previous versions.

---

## [Author Response]

Essential revisions:This seems like an excellent software package that could be further improved through the following additions:1. The validation looks impressive but is unfortunately qualitative throughout. I would like to see the appropriate statistics included for all of the relevant contrasts (e.g., Figures 1b-d, Pg. 12 final paragraph, etc.) used to validate these methods throughout the manuscript.

Many thanks for the reviewer’s feedback and support of the work. We should note, first, that most of our evaluations are quantitative (e.g. Figure 1B-D, Figure 2C-D, etc.). We have updated the manuscript in several locations to emphasize this and to include more statistical support.

These include:

– In Figure 1B and 1C, the addition of asterisks indicating statistically significant differences (paired t-test P<0.05) of the F1 score and Bray-Curtis similarity of MetaPhlAn 3 compared with all the other tools (as reported in the caption).

– In Figure 1D (and the related Figure S3), similar updates to indicate when HUMAnN 3’s community-level performance was significantly better than the other two methods in the evaluation (“*” implies *p* < 0.05 for all between-method paired *t*-tests). HUMAnN 3 did not detect significantly more species compared with HUMAnN 2 among skin and urogenital samples (all other comparisons were significant).

– In Figure 2A, we compared each dataset to the others using a PERMAVOVA test; all the p-values were statistically significant (P=0.001) as reported in the caption.

– In Figure 2D and 2F, we now highlight that, compared to the meta-analysis performed using MetaPhlAn 2 profiles that found 61 significantly associated species, the new analysis performed using MetaPhlAn 3 was able to identify 121 differential species under the same conditions.

2. It remains unclear why the performance of these different tools has been improved. Multiple ideas are mentioned in the text, including a more extensive database, improvements to parts of the algorithms, etc. The paper and field would benefit from a more targeted analysis to test what aspect of the update mattered most or alternatively if the improved performance is the aggregate result of a lot of little changes. At a minimum, it seems important to test how much these improvements are due to the expanded database as opposed to the tool itself. Could you get the same result with the version 2 tools and the up-to-date set of [meta]genomes?

To the extent that this is possible, we have included related results in the manuscript (e.g. individual HUMAnN 3 components in Supplementary Figure 4-5). However, this is a difficult comparison to define precisely in the general case, in large part since bioBakery 3 completely redefines its underlying gene catalog (ChocoPhlAn) to leverage UniRef and to accommodate many more available reference genomes. This means that there is no single way to run the new system with the old database, for example (e.g. some of the UniRef information on which the new pipeline is running are not available for the database on which MetaPhlAn 2 is based on), and running the old ChocoPhlAn pipeline on the new database is computationally infeasible.

However, we expect that overall performance improvements are due to both the expanded database and to the aggregate of other algorithmic changes, with a slight majority likely from the former. We now report the amount of species used in the synthetic metagenomes that could not be identified in MetaPhlAn 2 due to the absence of such species in its database, and this should be useful to put the improvements into the perspective of the expansion of the dataset:

“Of note, 379 of the total 1,119 (33%) species in the synthetic metagenomes were not present in the database of MetaPhlAn 2, emphasizing the role of expanded isolate genome availability in the improved detection capabilities of MetaPhlAn 3.”

We also added the following sentence into the Discussion:

“While the improvements were in large part a consequence of the much larger database of reference genomes that the system can now handle, additional algorithmic changes (Supplementary Table 2) were instrumental to provide more complete reporting and higher accuracy for references already available in previous releases.”

In the case of HUMAnN, we were able to more directly compare the impact of the expanded bioBakery 3 databases versus algorithmic improvements between v2 and v3 on accuracy and performance. This information was originally embedded in Figure S4 (HUMAnN parameter tuning) but was not clearly spelled out in the text. We have revised Figure S4 to specifically highlight HUMAnN’s accuracy and performance when running using v2 settings (vertical red lines) vs. the new defaults selected for v3 (vertical blue lines):

Improvements derived from algorithmic updates tended to be small in comparison with improvements from the larger databases, but they are not trivial: for example, relaxing the homology threshold used during translated search while simultaneously considering fewer, suboptimal hits allows us to map a considerably larger fraction of reads with essentially no drawbacks. We have revised the text to better emphasize this information:

“This difference [in specificity] is attributable in part to HUMAnN’s use of databasesequence coverage filters to reduce false positives, an approach introduced for translated search in HUMAnN 2 and expanded to nucleotide search in HUMAnN 3 (one of a number of algorithmic refinements in HUMAnN 3 that contribute to improved accuracy and performance even when controlling for database completeness; see Methods and Figure S4).”

“We performed a round of additional accuracy and performance tuning on these new parameters prior to the main evaluations of the paper (Figure S4) [Given that] these tuning evaluations include parameter configurations that are equivalent to HUMAnN 2 defaults applied to the expanded bioBakery 3 databases, they quantify improvements to HUMAnN 3’s accuracy and performance that are not attributable to its more complete database.”

“Supplementary Figure 4: Re-optimization of HUMAnN 3 based on the synphlan-humanoid metagenome and UniRef90 gold standard. HUMAnN’s accuracy and performance using v2 settings on v3 databases are highlighted with red vertical lines; changes in v3 are highlighted with blue lines.”

MetaPhlan 3 has clearly improved precision compared to the 2nd versions. It would be useful to know to what degree this is solely due to the updated ChocoPhlan database. A couple of algorithm improvements are mentioned. This would be possible with the introduction of "Metaphlan 3 with db 2 (or 2.7) ", at least in a subset of the comparisons in displayed in Figure 1? This should answer what role the algorithmic improvements played.

This is addressed in our previous response (#2); briefly, it is not possible either to run v2 with the v3 database (due to computational tractability) or v3 with the v2 database (due to changes in pangenome and marker definitions and formatting).

3. The figures and legends could use some polishing to enable clarity. For example, Figure 1d shows multiple bars/box plots, but it's unclear what they represent due to the lack of x axis labels. Figure 2a includes lots of different colors indicating study that are impossible to distinguish due to the massive number of red samples, the legend to Figure 2c is missing, and the countries in Figure 4a are impossible to see any pattern.

Apologies for the lack of clarity of the figures. We have added an x-axis label to Figure 1D to show that the panels depict sets of 5 metagenome samples analyzed using 3 different methods (boxplots indicate per-species accuracy within each of the 5 samples).

Figure 2A was intended to show a lack of separation between the studies considered in the meta-analysis. In order to make it more readable, we re-drew the figure by plotting the points in a random order, which prevents any one color from obscuring the others. For each dataset, we compared it to the others using a PERMAVOVA test; all the p-values resulted statistically significant (P=0.001). We included the missing legend to Figure 2C and better explained in the caption what the p-value is assessing.

For Figure 4A, the main result is the clustering of Chinese samples enriched in strains classified as Cluster 1, in addition to the non-geographically-specific subclade Cluster 2. To emphasize this, we have simplified the figure by merging multiple countries to continents.

4. The use cases of these updated tools are underwhelming and not clearly compared to the prior literature. The CRC analysis shown in Figure 2 highlight multiple bacterial species, one α diversity metric, and one gene (cutC) distinct from the healthy controls in this meta-analysis. It remains unclear which of these findings are new and if they are new, whether or not the new findings are better or worse than what was previously published. Similar issues affect the re-analysis of the IBD data (Figure 3) and the pan-genome data (Figure 4). While these figures provide a nice example of the analyses that can be done, it's unclear if anything new has been learned and if bioBakery 3 was necessary to run these analyses. These analyses also take up the majority of the main figures (3 out of 4), distracting from the main goal of the paper, which is to explain the improvements that have been made and to compare these updated tools to their previous versions.

While we agree that the first goal of the work is on reporting the performance of the tools, we also think that it is important to show that the new methods are able to (a) be readily applied on diverse, realistically-scoped “real world” data, and (b) that the re-analysis of existing data using the improved methods provides new biological information. To this end, and as suggested by the reviewers, we now better highlight what are the novelty points of potential biomedical relevance of our analyses:

“Compared to MetaPhlAn 2 when run on the same data, we found a total of 60 additional species that reached significance in the meta-analysis, confirming that the expansion of the database leads to improved biomarker discovery analysis.”

“[Using HUMAnN 3] we identified 558 ECs whose residual expression was significantly different (FDR q<0.05) in active CD compared with inactive CD (Figure 3B): a 66% increase compared to an identical analysis incorporating EC abundance profiles generated by HUMAnN 2 (Figure S11).”

“As these methods unravelled the population genomic structure of *R. bromii* that was not previously known, they can be similarly applied to hundreds of other host-associated or environmental microbial species to uncover their phylogenetic, functional, and transmission characteristics.”

While we are unable to include extensive text on these new findings in this manuscript due to space limitations – and they are in several cases the subject of their own, separate manuscripts in preparation – briefly, their significance (for CRC, IBD, and *R. bromii* phylogenetics, respectively) includes:

– We identified 60 species newly associated with CRC, in particular several strongly associated with CRC (*Dialister pneumosintes*, *Ruthenibacterium lactatiformans*, and *Eisenbergiella tayi*) that were previously not detectable by MetaPhlAn 2. These new species were also consistent with previously described patterns of greater richness and oral-type microbes in CRC-associated microbiomes.

– By re-analyzing the IBDMDB cohort with HUMAnN3, we identified 558 ECs significantly differentially expressed in active CD compared with inactive CD. We also identified under-expressed functions, such as galactonate dehydratase, which was highly encoded by *Escherichia coli,* but lowly expressed in active CD. *Hungatella hathewayi*, a new species included in version 3.0, is responsible for higher expression of the betaine reductase function from a small pool of genes.

– From the meta-analysis of 7,783 gut metagenomes, we identified two new phylogenetically, genomically, and biogeographically distinct subclades of *Ruminococcus bromii*, the first composed of strains retrieved from Chinese subjects and cohorts with rural and pre-industrial lifestyle and diet. These subclades are characterized by a higher genomic diversity and an enrichment of poorly characterized membrane proteins.